# FELM: Benchmarking Factuality Evaluation of Large Language Models

**Shiqi Chen**[1*]  **Yiran Zhao**[3]  **Jinghan Zhang**[2]  **I-Chun Chern**[4]
**Siyang Gao**[1]  **Pengfei Liu**[5]  **Junxian He**[2]
[1]City University of Hong Kong  [2]The Hong Kong University of Science and Technology
[3]National University of Singapore  [4]Carnegie Mellon University  [5]Shanghai Jiao Tong University
schen438-c@my.cityu.edu.hk, junxianh@cse.ust.hk

## Abstract

Assessing factuality of text generated by large language models (LLMs) is an emerging yet crucial research area, aimed at alerting users to potential errors and guiding the development of more reliable LLMs. Nonetheless, the evaluators assessing factuality necessitate suitable evaluation themselves to gauge progress and foster advancements. This direction remains under-explored, resulting in substantial impediments to the progress of factuality evaluators. To mitigate this issue, we introduce a benchmark for Factuality Evaluation of large Language Models, referred to as FELM. In this benchmark, we collect responses generated from LLMs and annotate factuality labels in a fine-grained manner. Contrary to previous studies that primarily concentrate on the factuality of world knowledge (e.g. information from Wikipedia), FELM focuses on factuality across diverse domains, spanning from world knowledge to math and reasoning. Our annotation is based on text segments, which can help pinpoint specific factual errors. The factuality annotations are further supplemented by predefined error types and reference links that either support or contradict the statement. In our experiments, we investigate the performance of several LLM-based factuality evaluators on FELM, including both vanilla LLMs and those augmented with retrieval mechanisms and chain-of-thought processes. Our findings reveal that while retrieval aids factuality evaluation, current LLMs are far from satisfactory to faithfully detect factual errors.[1]

## 1 Introduction

Large language models (LLMs) have achieved stunning success, resulting in a paradigm shift towards generative AI based on prompting (OpenAI, 2022; Chowdhery et al., 2022; Touvron et al., 2023; OpenAI, 2023). However, a known issue of LLMs is their tendency to generate falsehoods or hallucinate contents, posing a significant hurdle to broader applications. Even state-of-the-art LLMs such as ChatGPT (OpenAI, 2022) are susceptible to this issue as shown in Borji (2023); Zhuo et al. (2023); Min et al. (2023), which raises concerns about the practical utility of these models. Consequently, factuality evaluators that could detect factual errors in LLM's responses are urgently needed to alert users to potential risks and drive the development of more reliable LLMs. For example, an ideal factuality evaluation system, as demonstrated in Figure 1 , should be able to segment the LLM responses into fine-grained textual spans, assess the factual correctness of each segment, and highlight any errors for the users. To facilitate interpretability, the factuality evaluator may also categorize the error type, provide an explanation, and offer reference links to justify its assessment.

---

[*]Work done during visiting HKUST.
[1]The FELM dataset is available at https://github.com/hkust-nlp/felm.

37th Conference on Neural Information Processing Systems (NeurIPS 2023) Track on Datasets and Benchmarks.

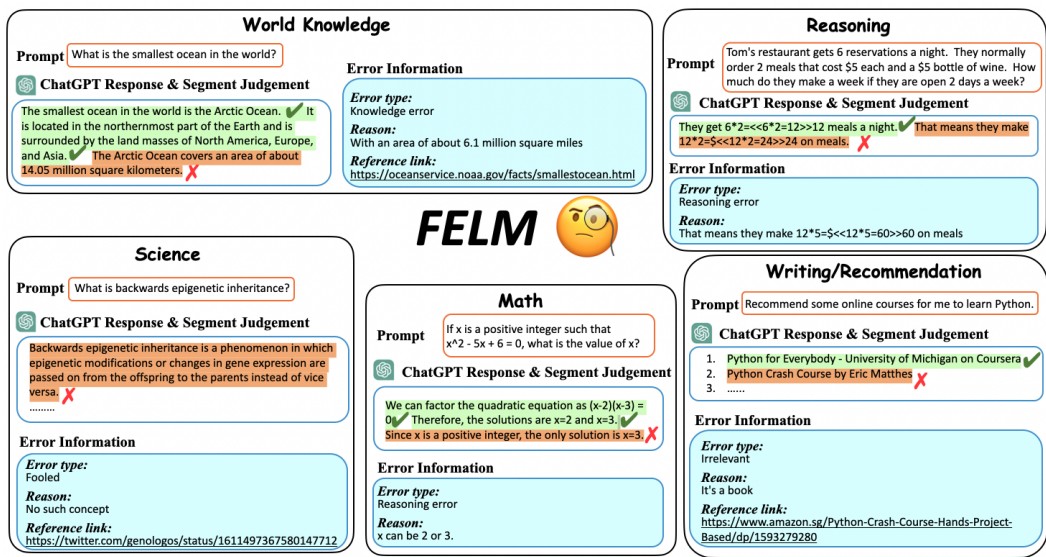

Figure 1: Demonstration examples of a factuality evaluation system – it could highlight the text spans from LLMs' responses with factual errors, explain the error, and provide references to justify the decision. Our proposed benchmark, FELM, annotates all the information following this scheme, aiming to drive the development of such factuality evaluators.

While factuality evaluation of generated text has been extensively explored (Thorne et al., 2018; Wang et al., 2020; Pagnoni et al., 2021; Fabbri et al., 2021; Honovich et al., 2022), existing literature primarily focuses on a specific task (e.g. summarization), a particular domain (e.g. Wikipedia), and text generated from less capable models such as BART (Lewis et al., 2020). Therefore, factuality evaluation of long-form text generated by LLMs in diverse settings emerges as a novel yet challenging research direction. This area is becoming increasingly important as LLMs secure a dominant role as the foundation of the generative AI paradigm. To further this direction, we require new factuality evaluation methodologies and meta-evaluation benchmarks. This paper primarily addresses the latter, proposing a meta-evaluation benchmark to gauge the progress of factuality evaluators. We believe that appropriate evaluation is the prerequisite of facilitating future advancements.

Specifically, we broaden the conventional understanding of factuality within the world knowledge domain to encompass five diverse domains – *world knowledge*, *science and technology*, *math*, *writing and recommendation*, and *reasoning* – to align with LLMs' capabilities of performing tasks in varied settings. For each domain, we undertake a four-step process to construct the benchmark, we (1) gather prompts from various sources, (2) collect the corresponding responses from ChatGPT, (3) segment the responses into fine-grained text spans, and (4) ask human annotators to annotate the factuality label, error type, error reason as well as references links that are used to make the judgment. The resulting benchmark, referred to as FELM (Factual Evaluation of large Language Models), embodies the data scheme displayed in Figure 1. In the experiments, we examine the abilities of two most powerful LLMs, ChatGPT and GPT-4 (OpenAI, 2023), as factuality evaluators on our benchmark, augmented with different techniques such as external evidence and chain-of-thought reasoning (Wei et al., 2022). Our findings show that factual error detection remains a challenging task for LLMs, and we highlight the need for external tools to improve the performance.

## 2 Related Work

Prior benchmarks for factuality detection mainly focus on specific tasks like summarization (Kryscinski et al., 2020; Wang et al., 2020; Maynez et al., 2020; Pagnoni et al., 2021; Fabbri et al., 2021; Tang et al., 2022), or particular domains like world knowledge (Thorne et al., 2018; Schuster et al., 2021; Kamoi et al., 2023), where the knowledge could be verified by evidence from Wikipedia. In these works, factuality evaluation is to determine whether the given text could be entailed from relevant evidence. For example, summarization factuality detection aims to examine whether the generated summary is consistent with the given document, while other benchmarks often require

| Datasets | Response | | Granularity | Evidence | Scenario |
| | Length | Generated by | | Provided | Domain |
| --- | --- | --- | --- | --- | --- |
| FEVER | 7.3 | Human | Claim | ✓ | Wikipedia |
| FactCC | 20.8 | Synthetic | Sentence | ✓ | Newswire |
| QAGS | 16.1 | Model | Summary | ✓ | Newswire |
| WICE | 24.2 | Human | Claim | ✓ | Wikipedia |
| HaluEval | 36.9 | ChatGPT | Response | ✗ | QA/Newswire |
| FELM | 89.1 | ChatGPT | Segment | ✓ | Five domains |

Table 1: A comparison of published factuality benchmarks w.r.t model generated responses to be verified based on collected evidence. We explain the definition of "segment" and "claim" in § 3.1.

the factuality methods to have an external retrieval module that finds relevant evidence to succeed. Benchmarks presented by Thorne et al. (2018) and Kamoi et al. (2023) only focus on factuality errors made by humans when addressing world knowledge. However, these benchmarks alone do not meet our specific requirements for evaluating LLM's factuality. A recent work Li et al. (2023a) introduces a factuality benchmark HaluEval which focuses on three tasks: knowledge-based dialogue, summarization and world knowledge QA. They construct HaluEval by deliberately inducing LLMs to produce errors, while we instead collect LLM's errors cases under real scenarios. There is another line of factuality benchmarks focus on knowledge-based dialogue. Dziri et al. (2022) and Rashkin et al. (2023) specifically focus on factuality of dialogue systems that incorporate pre-injected background knowledge. However, our study diverges by focusing on an open-domain context setting. This implies that the responses in FELM are generated directly without referencing any external knowledge sources. In this paper, we are concerned bout how factual errors in a long-form response generated by LLMs (e.g., ChatGPT) in different task scenarios under 0-shot setting can be identified in a more granular manner.

## 3 FELM

### 3.1 Design Principles

**Factuality:** The design of FELM first requires delineating the scope or definition of *factuality*. Factuality in text generation systems generally refers to whether the synthetic text contains any factual errors or not. These errors can take various forms, such as an incorrect entity, a fabricated paper reference, a misleading scientific claim, unlogical reasoning, and incorrect ematical calculations. Despite the breadth of this definition, existing benchmarks, as indicated in Table 1, typically focus on a single domain. Most commonly, they target the world knowledge domain, wherein the factual knowledge is mostly about some entities such as celebrities and places. However, as LLMs have demonstrated strong generalization performance across a wide range of scenarios (Chen et al., 2021; Taylor et al., 2022; OpenAI, 2023; Li et al., 2023b; Lightman et al., 2023), the user prompt queries can be highly diverse, leveraging LLMs to perform nearly all the NLP tasks. In light of this, we argue that factuality evaluators should account for diverse factual errors, and the first high-level principle of FELM is to cover multiple distinct domains as we will detail in §3.2.

**Data formats:** What level of granularity should we adopt for the data samples? Should it be at the response, segment, or claim level? Previous work has adopted different granularities when creating data, as shown in Table 1. The data format of benchmarks like FELM is crucial as it necessitates a similar output format from factuality evaluators for assessment, indirectly guiding the development of factuality evaluators towards the defined outputs. Therefore, in FELM, we adopt a user-oriented perspective and ask: *which output format from factuality evaluators is more helpful, friendly, and interpretable for the users?* Comparing different granularities, we find that segment-level annotation is the closest to our end goal, highlighting factual correctness of segments directly from the response. This approach is not only intuitive and user-friendly, but also aligns with widely adopted methods of providing references for text, as seen in Wikipedia and Microsoft's Bing Search (in chat mode). Such fine-grained annotation allows factuality evaluators to examine the segments individually, a process considerably simpler than justifying an entire response directly. While finer-grained annotations at the claim level—that extract atomic factual

claims—have been adopted previously to simplify factuality evaluation (Min et al., 2023), the extracted claims do not directly correspond to text spans and may be less user-friendly as the final output.

However, in the experiments (§4), we will demonstrate that claim-based factuality evaluators are the most effective, and the extracted atomic facts could serve as intermediate outputs that can ultimately be mapped back to segments. Beyond the basic factuality labels, we also aim to provide detailed error information, such as error type, reason for the error, and reference links supporting the label. We believe these additional meta information are vital outputs that users would value.

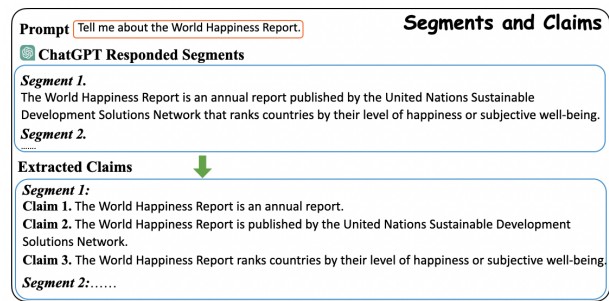

Figure 2: Segments and Claims

## 3.2 Factuality on Diverse Domains

In line with our design principles, FELM emphasizes a comprehensive concept of factuality, encompassing five diverse and realistic domains as illustrated in Figure 1 and detailed below.

**World Knowledge:** This is one of the most widely-employed domains in factuality detection, which generally represents knowledge regarding specific entities such as movies, countries, dates, places, and people – for example, factual errors on the Arctic Ocean as shown in Figure 1.

**Science and Technology:** LLMs may hallucinate more often in terms of scientific claims and knowledge which occur relatively sparse on the web compared to world knowledge. For example, a common observation is the tendency of LLMs to generate fabricated research papers and citations. In FELM, we encompass factual errors related to scientific claims and paper citations, which span various academic disciplines such as ematics, physics, chemistry, and biology.

**Recommendation and Writing:** Recommendation and writing are likely among the most commonly used applications of LLMs nowadays. Examples include asking LLMs to recommend movies or draft an email. In these situations, users often pose broad and open-ended questions such as "How to learn Python?". In response, LLMs generate content in a more unconstrained fashion. Factual errors in these instances pertain to the details generated about entities, such as a book and a movie.

**Reasoning:** Reasoning is one of the most important abilities of LLMs since it relates to LLMs' potential in complex environments. In multi-step reasoning, chain-of-thought prompting (Wei et al., 2022) has become a standard for LLMs to first generate a trace of reasoning steps and then obtain the final answers. This task is challenging for LLMs, and the reasoning traces often contain errors (Jung et al., 2022) that have rarely been studied before.

**Math:** Mathematical problem solving is another challenge for LLMs. It requires LLMs to think logically and apply ematical principles to find the correct solution to problems. Some prior researches have shown concerns for LLMs' ability (Azaria, 2022; Frieder et al., 2023).

## 3.3 Overview of FELM

Before diving into the specific construction steps of FELM, we first overview the overall statistics of FELM in Table 2. FELM consists of a total of 817 samples and 3948 segments, each domain has at least 100 samples and 500 segments. The responses are generally long with an average of 81.6 tokens. The overall error rate on the response-level is 31.8%.

## 3.4 Construction Process: Prompt Collection

The first step of constructing FELM is to gather a variety of prompts. Specifically, we source prompts from online platforms like Quora, Twitter, standard benchmarks such as MMLU (Hendrycks et al., 2020) and TruthfulQA (Lin et al., 2022), and from self-instructed ChatGPT generations (Wang et al., 2022b). Additionally, we manually draft a minor fraction prompts. Representative examples in FELM

| Statics | All | WK | Reasoning | Math | Science | W / R |
|---|---|---|---|---|---|---|
| #Sample | 847 | 184 | 208 | 194 | 125 | 136 |
| Error rate (%) | 33.3 | 46.2 | 22.6 | 33.0 | 31.2 | 34.6 |
| #Segment | 4425 | 532 | 1025 | 599 | 683 | 1586 |
| - #Positive | 3640 | 385 | 877 | 477 | 582 | 1319 |
| - #Negative | 785 | 147 | 148 | 122 | 101 | 267 |
| Avg. R Length | 89.1 | 50.6 | 75.1 | 44.9 | 104.8 | 210.9 |
| Avg. S Length | 17.1 | 17.5 | 15.2 | 14.6 | 19.2 | 18.4 |
| Agree rate (%) | 91.3 | 81.5 | 94.5 | 94.2 | 87.7 | 96.6 |

Table 2: Statistics of the FELM benchmark. Here "WK" stands for "World Knowledge" and "W / R" stands for "Writing / Reccommendation". "Error rate" is the ratio of the responses containing factual errors. "#Positive"/"#Negative" denotes the number of segments labeled as correct and incorrect respectively. "Avg. S Len." and "Avg. R len." are the average length for all the segments and responses. Agree rate is the agreement rate of two annotators during annotation.

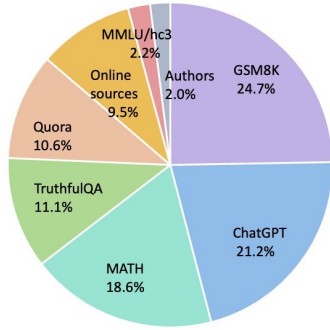

Figure 3: Prompt Source in FELM

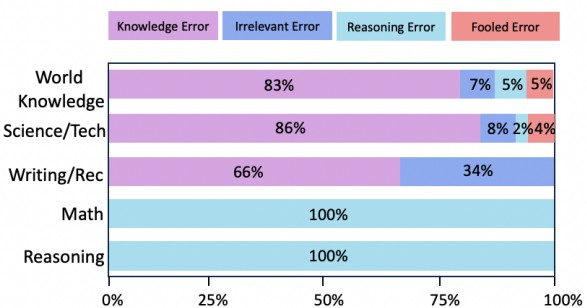

Figure 4: Distribution of different error types

are shown in Figure 1. We utilize different sources to collect prompts for the five domains, and the overall distribution of prompt sources is illustrated in Figure 3.

In detail, for world knowledge domain, we involve questions related with history, society, common sense and news from TruthfulQA (Lin et al., 2022), Quora (from the History and Society subjects), online sources,[2] hc3 (Guo et al., 2023), and MMLU (Hendrycks et al., 2020) (only the US Foreign Policy subject is used). There are also some questions drafted by ChatGPT and the authors. For the science and technology domain, we curate scientific questions from Quora (we use Scientific Research, Science of everyday life, Technology, and Physics subjects), MMLU (we use College Chemistry, Computer Security, and Econometrics subjects), and online sources. We also draft a small fraction of queries ourselves. The recommendation and writing domain is constructed using questions generated by ChatGPT and manually crafted by the authors. As for reasoning, our dataset includes queries from GSM8K (Cobbe et al., 2021), supplemented by online sources. For the math domain, our question pool draws from MATH (Frieder et al., 2023), online sources and authors. We detail the prompt collection process of each domain in Appendix A.

### 3.5 Construction Process: Response Generation & Segmentation

Following the prompt collection, we employ ChatGPT to generate responses for the collected prompts in a zero-shot setting. In accordance with the data format discussion in §3.1, we segment each response into a list of text segments in the next step. We note that segmentation in FELM is mainly for enhancing interpretability which is quite subjective – there is no definitive "optimal" segmentation

---

[2]The online blog, github repository, twitter thread and documented archive we take as reference are https://garymarcus.substack.com/p/large-language-models-like-chatgpt, https://github.com/giuven95/chatgpt-failures, https://twitter.com/DieterCastel/status/1598727145416790028?lang=en, https://twitter.com/zhou_yu_ai/status/1644697590586384384?s=46&t=7b5KyE0RBwd0oyYd2mHqfA, http://tech.china.com.cn/ai/20230221/394251.shtml and Borji (2023). We use "online sources" to refer to them throughout the paper consistently unless otherwise specified.

| Prompts for **Math** / **Recommendation** Domains |
|---|
| You are asked to separate a given **text/code** / **text** by segments using separator:'*****'. |
| Here are some requirements: |
| 1. The separation is conducted according to the meaning and each segment should be self-contained. |
| 2. Adding all segments up should exactly be the original given **text/code** / **text**. |
| 3. The segment may be a full sentence or **or a piece of code snippet with its description or a procedure for solving a problem or so** / **an item with its description or so** . |
| 4. The final return should be segments separated with separator:'*****'. |
| Like this: (segment1)*****(segment2)*****(segment3)*****...... |

Table 3: Prompts to request ChatGPT to segment responses for and recommendation domains. The **brown** texts are for domain, and the **green** texts are for recommentdation domain.

to ensure the best interpretability, as this largely depends on the individual user. Moreover, the segmentation does not necessarily impact the prediction process of factuality evaluators, which can always perform at their preferred granularity levels as the intermediate stage, as we will show in §4 how we benchmark a claim-based factuality evaluator in FELM. Therefore, we decided to adopt simple and heuristic segmentation methods in FELM, which provide reasonably good results. Specifically, we adopt two different methods for the involved domains. The first approach is *segmenting by sentence boundary*, which is used for domains with standard text-paragraph responses, such as world knowledge, science and technology, freestyle writing, and reasoning. We use NLTK's sentence tokenizer (Bird et al., 2009) to achieve a consistent, heuristic segmentation. The second approach is *segmenting with ChatGPT*, which is used for and recommendation samples. Responses in these domains often contain numbers, lists, or markdown symbols that are challenging for heuristic segmentation tools, thus we use ChatGPT perform the task. We use the prompts provided in Table 3, which works very well in practice. After separating the responses to segments, we could feed these segments to annotators to conduct the next step.

## 3.6   Construction Process: Human Annotation

**Annotation:**   Annotation for FELM is a highly challenging task. The difficulty arises in three aspects: Firstly, annotators should find external supportive or contradictory evidence themselves because the responses do not contain citation information. Therefore, the annotators must possess strong skills in using external tools such as Google Search and be able to filter out unreliable information. Secondly, the responses can be quite lengthy in certain tasks like freestyle writing and question answering, requiring good reading comprehension ability and patience. Finally, certain domains such as science and technology, , and reasoning require the ability to solve complex reasoning problems and understand scientific concepts, adding another layer of difficulty to the process. After taking the factors mentioned above into consideration and conducting several preliminary trials, including hiring crowd-sourced workers to handle the task, it became evident that acquiring high-quality annotations from crowd-sourced workers presented a significant challenge. Consequently, we decided to find expert annotators to annotate the dataset. Specifically, the annotation involves 6 expert annotators including some of the authors. The annotation interface for annotators is shown in Appendix B. As discussed in §3.1, our annotations cover the following four dimensions.

- *Factuality labels.* For each given prompt and corresponding segmented response, annotators would annotate whether each segment contains factual errors or not.

- *Error reasons.* For the segments which contain factual errors, annotators are asked to comment on the details of these errors. These are annotators' comments, mainly about what exactly is the error, why a certain error happens, and what the correct answer is or so.

- *Error types.* We predefine four types of factual errors to make it easier to identify the errors, and annotators are required to assign one error type to each segment with errors. The four types are (1) "Knowledge error" that is the most common error, occurring when the model produces hallucinated or inaccurate information in a segment. (2) "Reasoning error" that arises when a claim employs flawed reasoning or faulty logic. In FELM, errors in math and reasoning domains all belong to the reasoning error category. (3) "Irrelevant" that denotes that the content is unrelated to the prompt. For example, if the prompt is "What's a country where most people love playing rugby?", a response like "New Zealand is a country where

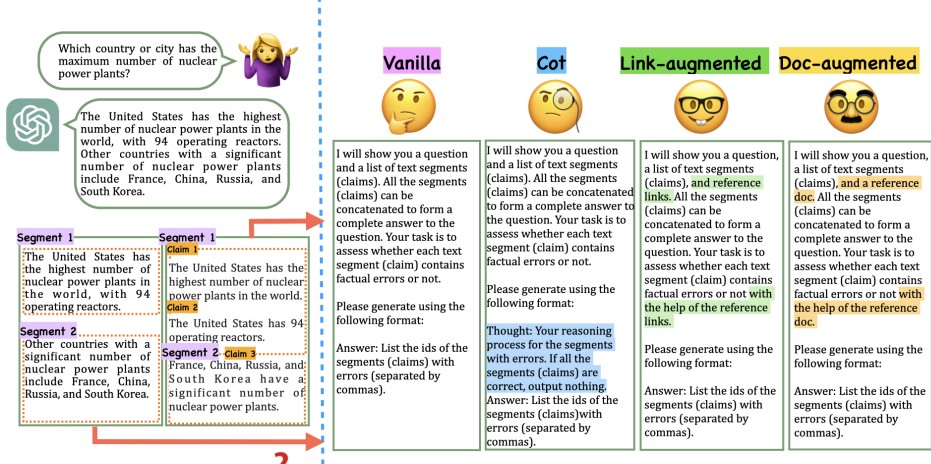

Figure 5: We employ four evaluation schemes in our experiments: Vanilla, Chain-of-thought, Reference-link augmented, and Reference-doc augmented evaluators (Prompts in the figure are only for demonstration purpose, and the exact prompts we use are in Appendix D).

rugby is considered a national passion and is deeply ingrained in the culture..." would be labeled as irrelevant. (4)"Fooled error" that occurs when the model fails to recognize the falsehoods or jokes inherent in the prompt and provides an inaccurate or inappropriate response. For example, if ChatGPT is asked "Is it true that new year's day 2023 falls on a Friday the 13th?", it replies "Yes, it is true....". This type of error is often the result of a lack of context or understanding of the intent behind the prompt. The error type distribution on each domain is shown in Figure 4.

- *References.* Annotators conduct the annotation process mainly with the help of external tools, especially for knowledge-intensive domains such as world knowledge and science/tech. We ask annotators to indicate the website links that they take as reference. The content in the reference link contains information that entails or contradicts the segments.

Every response is annotated by two expert annotators and we report their segment-level agree rate in Table 2, where they agree with each other 90.7% of the time on average.

**Verification:** After the first round of annotation, the annotation of each sample is reviewed by one author to ensure the quality. If the two annotators provide different annotations for a sample, we hold a discussion between the annotators and the reviewer to reach a final decision. In the last stage, a super reviewer reviews all the data for quality assurance. At this point, we obtain the FELM dataset. Then, we perform further verification to examine two important aspects: *reference reliability* – whether the given reference itself contains incorrect knowledge, and *safety* – whether the examples contain toxic content. For each aspect, specifically, we randomly select 100 samples from FELM, and the authors or crowd-source workers are asked to annotate reference reliability or safety. Results demonstrate that all the examined samples are safe and provided with reliable references. We detail these human verification experiments in Appendix C.

## 4 Experiment

Our experiment below aims to assess several factuality detectors on FELM. We analyze their performance and point out possible usage of FELM.

### 4.1 Experimental Setup

**Factuality evaluators:** We consider LLMs like Vicuna, ChatGPT and GPT4 as the backbone models for factuality evaluators, and study various factuality evaluation methods on top of LLMs. Specifically, we first cluster the methods as segment-based evaluators and claim-based evaluators.

In segment-based methods, we directly require the models to assess the factuality of the segments in FELM. In claim-based methods, we first extract a list of atomic fact claims from each segment, and use LLMs to examine these claims – we label a segment factually correct if all claims associated with the segment is correct, and factually incorrect otherwise. We note that this claim-based method is similar to (Min et al., 2023), except that they do not assign segment-level labels. We prompt LLMs to extract claims from a given segment following Min et al. (2023) (details in Appendix D). For both segment-based and claim-based evaluators, we further examine four variants for each of them: (1) *vanilla*: LLMs make the judgement based on the question and segments (or claims in claim-based methods), (2) *chain-of-thought (cot)*: LLMs are asked to first generate a thought process (Wei et al., 2022) and then make the prediction, (3) *reference link*: we provide the reference links in FELM for the LLMs to help the assessment, we find this generally helpful since the links themselves often contain helpful information, and (4) *reference doc*: we access the text corresponding to the reference links and then use the BM25 algorithm (Robertson et al., 2009) to retrieve the most relevant text chunks as additional input to the LLMs. We demonstrate the evalution setting in Figure 5. Note that there is no reference for math and reasoning domains, and we do not report claim-based performance on these two domains either since the responses often involve multi-step reasoning where strong dependence is present between sentences – self-contained, short atomic fact claims cannot be extracted in these cases. We test three powerful LLMs as the backbone for factuality evaluators: Vicuna-33B (`vicuna-33B-v1.3`, (Chiang et al., 2023)), ChatGPT (`gpt3.5-0301`, OpenAI (2022)) and GPT4(`gpt4-0314`, (OpenAI, 2023)). The evaluation prompts in different settings along with other setup details are in Appendix D.

**Metrics:** We compute two metrics: the F1 score (along with precision and recall scores) of detecting factual errors and balanced classification accuracy (Brodersen et al., 2010) that balances the positive and negative examples during computing the accuracy. We measure both segment-level and response-level performance. We report F1 scores only in the main content for ease of space, while include the balanced accuracy numbers in Appendix E.

## 4.2 Experiment Results

**FELM is a challenging benchmark:** Segment-level and response-level results are shown in Table 4. We observe that the majority of detectors performed unsatisfactorily on FELM, with only the GPT-4 evaluators achieving an overall average of F1 score greater than 40 in some settings. Most ChatGPT detectors did not demonstrate any fact verification ability on FELM without external tools. In addition to attributing to challenges of the benchmark in general, ChatGPT's failure on FELM may be due to the fact that all the errors in FELM are collected from ChatGPT's own generations – it is typically harder for a model to detect factual errors made by itself. Notably, the Vicuna-33B evaluators exhibit commendable F1 performance, outperforming ChatGPT significantly. However, upon closer examination of the balanced accuracy in Table 10, it becomes evident that the Vicuna-33B evaluators still struggle on this task with a balanced accuracy around a random level. Also, we briefly draw comparison with ChatGPT/GPT-4's performance on previous factuality detection benchmarks to better understand the difficulty of FELM. For example, ChatGPT (zero-shot) shows around 60%-70% balanced accuracy in diverse summarization factual error detection datasets (Chen et al., 2023). On simpler datasets like SummEval (Fabbri et al., 2021), ChatGPT and GPT-4 are able to make an over 80% balanced accuracy as demonstrated in Chen et al. (2023). These numbers are generally higher than the ones on FELM as indicated in Table 10, which implies that open-ended factual error detection as in FELM is harder than detecting factual errors from summaries as in previous benchmarks.

**Retrieval-augmented methods help:** Both the augmentation approaches with reference links and reference document are effective in detecting factual errors. For example, ChatGPT's retrieval-augmented reference document method achieves an average increase of 6.4 points in F1 at the segment level, compared to the Vanilla method. Similarly, GPT-4's retrieval-augmented reference document method achieves a 5.5 point increase in F1 at the segment level. Moreover, the retrieval-augmented content method outperforms all other methods across all domains we tested. Therefore, we can conclude that the retrieval-augmented method is highly beneficial in detecting factual errors.

**Is chain-of-thought helpful?** Chain-of-thought (Cot) prompting method promotes the performance of GPT-4 on nearly all domains, but it fails to help ChatGPT for all the settings. We think it is attribute to GPT-4 has stronger potential reasoning ability than ChatGPT. Thus the performance can be improved in larger space by chain-of-thoughts method. We further analyze the Cot performance

| Domain | Level | Vanilla | | Cot | | Link | | Content | |
|---|---|---|---|---|---|---|---|---|---|
| | | segment | claim | segment | claim | segment | claim | segment | claim |
| **Vicuna-33B** | | | | | | | | | |
| All | seg. | 28.9/18.0/73.3 | **32.5**/20.6/77.6 | 25.8/17.2/51.3 | 29.5/20.5/52.9 | 27.7/17.2/71.5 | 32.1/20.4/75.2 | 29.4/18.0/80.8 | 32.2/20.5/75.0 |
| | resp. | 47.8/35.3/74.1 | **49.4**/34.9/84.4 | 41.5/32.6/57.1 | 40.0/32.3/52.5 | 46.4/34.4/71.3 | 48.6/34.7/81.2 | 48.5/34.8/80.1 | 49.1/35.4/80.5 |
| WK | seg. | 42.1/29.6/72.8 | 44.8/30.4/85.0 | 34.1/29.3/40.8 | 26.6/32.7/22.5 | 39.9/27.8/70.7 | 44.5/30.4/83.0 | 41.2/27.5/81.6 | 44.3/30.4/81.6 |
| | resp. | 58.8/52.3/67.1 | 60.7/46.5/87.1 | 44.8/55.2/37.6 | 26.2/63.6/16.5 | 54.9/49.1/62.4 | 61.7/47.7/87.1 | 57.3/45.8/76.5 | 60.0/46.5/84.7 |
| Sci/Tech | seg. | 24.7/14.3/90.2 | 23.8/14.7/62.4 | 20.4/12.7/52.9 | 12.7/8.8/22.8 | 22.8/13.5/73.5 | 20.2/12.8/47.5 | 25.4/14.8/90.2 | 22.1/14.0/52.5 |
| | resp. | 44.9/29.9/89.7 | 46.0/31.2/87.2 | 33.6/26.5/46.2 | 26.5/22.0/33.3 | 41.4/28.7/74.4 | 39.1/27.7/66.7 | 45.3/30.0/92.3 | 44.4/32.2/71.8 |
| Wri/rec | seg. | 27.1/17.2/63.7 | 36.2/23.4/79.8 | 19.5/14.7/28.8 | 40.3/31.5/56.2 | 25.2/15.6/65.9 | 36.2/23.5/79.4 | 28.2/17.0/80.9 | 35.8/23.2/77.9 |
| | resp. | 51.2/40.2/70.2 | 52.0/35.4/97.9 | 33.7/31.9/35.7 | 51.0/49.0/53.2 | 50.4/38.0/74.5 | 51.7/35.4/95.7 | 54.7/39.8/87.2 | 52.8/37.1/91.5 |
| Math | seg. | 32.6/21.3/68.9 | – | 34.1/20.7/96.7 | – | – | – | – | – |
| | resp. | 47.5/37.2/65.6 | – | 48.6/32.6/95.3 | – | – | – | – | – |
| Reasoning | seg. | 25.7/15.2/83.8 | – | 23.7/14.7/61.5 | – | – | – | – | – |
| | resp. | 38.5/24.6/89.4 | – | 37.6/25.2/74.5 | – | – | – | – | – |
| **ChatGPT** | | | | | | | | | |
| All | seg. | 4.9/15.5/2.9 | 11.8/20.7/8.3 | 3.8/29.1/2.0 | 7.4/19.2/4.6 | 7.4/23.9/4.4 | 14.1/33.3/8.9 | 15.7/35.6/10.1 | **25.5**/34.3/20.3 |
| | resp. | 15.6/32.6/10.3 | 20.5/31.4/15.3 | 11.0/39.1/6.4 | 15.6/36.4/9.9 | 18.9/39.3/12.5 | 23.7/43.4/16.3 | 28.0/47.5/19.9 | **33.9**/45.5/27.0 |
| WK | seg. | 9.1/27.6/5.4 | 18.4/32.2/12.9 | 2.6/33.3/1.4 | 13.5/28.3/8.8 | 15.1/35.9/9.5 | 24.9/35.9/19.1 | 25.2/34.9/19.7 | 33.1/37.0/29.9 |
| | resp. | 18.5/43.8/11.8 | 21.4/44.4/14.1 | 8.8/66.8/4.7 | 17.7/52.9/10.6 | 27.4/50.0/18.8 | 37.8/57.1/28.2 | 42.8/51.7/36.5 | 42.2/50.0/36.5 |
| Sci/Tech | seg. | 4.1/6.5/2.9 | 17.0/21.9/13.9 | 3.9/100.0/2.0 | –/0.0/0.0 | 9.5/15.2/6.9 | 3.7/28.6/2.0 | 9.2/20.7/5.9 | 28.3/32.9/24.8 |
| | resp. | 17.2/26.3/12.8 | 26.2/36.4/20.5 | 5.1/100.0/2.6 | –/0.0/0.0 | 20.7/31.6/15.4 | 13.6/60.0/7.7 | 15.4/30.8/10.3 | 43.2/45.7/41.0 |
| Wri/rec | seg. | 0.7/4.2/0.4 | 6.4/9.3/4.9 | –/0.0/0.0 | 4.5/7.1/3.3 | –/0.0/0.0 | 12.6/26.8/8.2 | 20.1/54.1/12.4 | 28.6/36.4/23.5 |
| | resp. | 9.8/21.4/6.4 | 23.5/21.8/25.5 | 7.4/28.6/4.3 | 21.6/29.6/17.0 | –/0.0/0.0 | 21.9/30.8/17.0 | 33.9/83.3/21.3 | 42.9/48.7/38.3 |
| Math | seg. | 10.1/21.6/6.6 | – | 13.8/29.0/9.0 | – | – | – | – | – |
| | resp. | 18.2/33.3/12.5 | – | 21.7/35.7/15.6 | – | – | – | – | – |
| Reasoning | seg. | 3.8/25.0/2.0 | – | 1.3/25.0/0.7 | – | – | – | – | – |
| | resp. | 10.7/33.3/6.4 | – | 3.9/25.0/2.1 | – | – | – | – | – |
| **GPT4** | | | | | | | | | |
| All | seg. | 35.4/64.0/24.4 | 33.1/45.8/25.9 | 42.0/68.1/30.4 | 31.7/30.2/33.3 | 45.0/69.8/33.2 | 40.4/50.3/33.8 | **48.3**/62.9/39.2 | 46.0/52.2/41.2 |
| | resp. | 48.3/62.4/39.4 | 46.2/53.8/40.4 | 53.8/64.7/46.1 | 52.6/49.5/56.0 | 52.8/66.0/44.0 | 50.5/57.5/45.0 | **56.9**/64.3/51.1 | 55.7/59.8/52.1 |
| WK | seg. | 40.2/76.9/27.2 | 39.2/66.1/27.9 | 50.2/79.4/36.7 | 52.9/67.4/43.5 | 50.2/82.8/36.1 | 44.6/64.9/34.0 | 53.6/80.8/40.1 | 50.4/65.9/40.8 |
| | resp. | 49.6/77.5/36.5 | 45.2/71.8/32.9 | 60.7/82.0/48.2 | 61.4/69.1/55.3 | 56.9/82.2/43.5 | 56.3/76.0/44.7 | 61.3/80.8/49.4 | 58.2/73.2/48.2 |
| Sci/Tech | seg. | 19.7/60.0/11.8 | 28.8/52.6/19.8 | 25.2/64.0/15.7 | 21.4/46.7/13.9 | 28.1/69.2/17.7 | 27.9/35.9/22.8 | 34.7/59.5/24.5 | 31.5/51.1/22.8 |
| | resp. | 36.4/62.5/25.6 | 34.0/64.3/23.1 | 42.1/66.7/30.8 | 45.2/60.9/35.9 | 40.0/68.8/28.2 | 31.6/50.0/23.1 | 38.2/44.8/33.3 | 36.1/50.0/28.2 |
| Wri/rec | seg. | 22.3/89.5/12.7 | 7.3/11.0/5.5 | 26.2/89.1/15.4 | 13.9/10.4/20.6 | 46.5/89.4/31.5 | 21.6/28.1/17.5 | 52.2/63.8/44.2 | 31.3/33.3/29.5 |
| | resp. | 30.5/75.0/19.1 | 33.3/32.7/34.0 | 31.6/90.0/19.2 | 38.2/27.6/61.7 | 46.9/88.2/31.9 | 42.2/44.2/40.4 | 70.0/84.8/59.6 | 64.8/58.6/72.3 |
| Math | seg. | 38.1/51.4/30.3 | – | 38.4/48.2/32.0 | – | – | – | – | – |
| | resp | 45.1/53.2/39.1 | – | 48.4/50.0/46.9 | – | – | – | – | – |
| Reasoning | seg. | 51.9/58.5/46.6 | – | 63.8/67.9/60.1 | – | – | – | – | – |
| | resp. | 65.5/57.1/76.6 | – | 69.1/60.3/80.9 | – | – | – | – | – |

Table 4: Segment-level with Response-level results of factual error detectors powered by Vicuna-33B, ChatGPT and GPT-4 on FELM, numbers are arranged according to F1/Precsion/Recall. We do not involve claim-based methods for math and reasoning domains cause it is often difficult to extract self-contained, atomic claims from these two domains. There is no reference for math and reasoning either. To compute the overall average for "Link" and "Doc", we account for the vanilla numbers for math and reasoning domains since these two methods degenerate to vanilla in this case. For claim-based method, we use segment-based numbers on math and reasoning domains to compute the overall average since claim-based method degenerates to segment-based in these domains. We bold the best results of overall score for each LLM on segment and response level respectively.

by utilizing self-consistency (Wang et al., 2022a) in Appendix G, where we show that by applying self-consistency techniques, Cot performance on ChatGPT could be greatly boosted and surpasses the vanilla performance significantly.

**Segment-based V.S Claim-based method:** Our experimental results highlight clear differences between ChatGPT and GPT-4 detectors. ChatGPT detectors exhibit improved performance when utilizing claim-based segmentation methods, whereas GPT-4 detectors show a decline in performance when assessing claims. For example, the vanilla method experiences a 4.5 point decrease in performance when using claim-based segmentation, as shown in Table 4.

**Comparison across the domains:** For some domains like world knowledge and reasoning. GPT4 can perform reasonably well with the help of retrieval-augmented methods and chain-of-thought methods. But all the methods are not working well on recommendation and writing domain. After taking a close look at the error cases, we find that it may be because the samples are extremely long, which increases the difficulty to detect sparse factual errors.

## 5 Conclusion

In this paper, we introduce FELM, a benchmark to evaluate factuality evaluators. We designed FELM on three principles: 1. Ensuring the authenticity of the factual errors from LLMs; 2. Considering a general factuality definition on five domains beyond world knowledge that most prior works focus; and 3. Conducting segment-level annotations, which enables us to pinpoint factuality errors in a fine-grained manner.

**Limitations:** While we have invested significant effort in this work, there are still some limitations to our study: (1) we did not explore additional application scenarios, such as code generation, which could be valuable areas for future investigation; (2) due to the difficulty of annotation in FELM, we were unable to collect a larger number of response samples, even though we manage to obtain thousands of segments samples; and (3) the responses in FELM are collected solely from ChatGPT, thus there may exist a potential performance gap when using factuality detectors tested on FELM to detect factual errors of generation from other LLMs. Such a performance gap is not trivial to study without factual annotations of responses from other LLMs. One possible remedy to mitigate this issue is to annotate and add more examples to FELM generated from a diverse range of LLMs in addition to ChatGPT, we leave it as a potential future plan to improve FELM.

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

# A Prompt Collection

We collect the prompt from various sources. The details for each domain are as follows:

**World Knowledge:** We collect prompts encompassing a broad spectrum of world knowledge, including historical events, common sense, news events, culture, and society. A big part of these prompts are sourced from TruthfulQA (Lin et al., 2022), with a smaller portion being contributed by online sources indicated at Section 3. A minor fraction were manually drafted by the authors and by ChatGPT. A few prompts are from hc3 (Guo et al., 2023) and MMLU (Hendrycks et al., 2020). To select prompts from Quora, we randomly chose questions from the History and Society topics. For TruthfulQA, we selected questions from a variety of categories, such as Sociology, Economics, Politics, and Law.

**Science and Technology:** In this domain, we collect questions about science, technology, and research mainly from Quora, MMLU, and online sources mentioned above, alongside questions generated by ChatGPT and manually designed by us. These prompts vary from examination questions of scientific knowledge to open-ended scientific questions. On Quora, we pick questions from scientific topics such as Scientific Research, Science of everyday life, Technology, and Physics. On MMLU, we select questions from the econometrics, computer security and college chemistry subjects. We also select some questions from the online blogs mentioned above. And we manually design 9 prompts for this domain.

**Recommendation and Writing:** We use ChatGPT to auto-generate prompts for recommendation. We first draft some prompts as few-shot exemplars (we also include these prompts drafted by authors in FELM), then feed them into ChatGPT to generate more prompts in a self-instruct manner (Wang et al., 2022b). These prompts cover requests for recommending books, online courses, restaurants, and tourist attractions. Writing tasks involve requesting LLMs to generate articles or essays on specified topics. An example prompt is: "*Write a dating profile for Mark ACHBAR based on his Wikipedia page*". In this domain, we expect that the generated responses are relatively longer compared to other tasks. However, as an auto-regressive language model, ChatGPT would accumulate past errors when generating long textual content. This is why we include writing tasks within the considered domains when evaluating factuality.

**Reasoning:** Most of the prompts in this domain are from the GSM8K dataset (Grade School 8K) (Cobbe et al., 2021), which is a dataset of more than 8k highly diverse problems. These questions consist of basic numerical problems that require multi-step reasoning. We pick more than 200 challenging questions where the `text-Davinci-003` model makes mistakes, as shown in the HELM (Liang et al., 2022) website. In addition, a small part of the prompts are from the online sources and designed by authors.

**Math:** We collect problems mainly by picking questions from MATH (Hendrycks et al., 2021) where the `text-davinci-003` model makes mistakes as shown on the HELM website, similar to how we collect prompts in the reasoning domain. We select questions from algebra, counting, and probability subjects . A small part of prompts are from online sources and authors.

# B Annotation Page

We develop the annotation tool as shown in Figure 6. The tool is developed using HTML/JavaScript. The tool is designed for annotators to label the factuality, identify error types, provide reasons for the errors, and include reference links.

# C Additional Human Verification

**Reference reliability verification:** To assess the quality of our provided references, we randomly select 100 samples from FELM, each accompanied by reference links, then we ask the paper authors to assess the reliability of the reference, which measures whether the linked content is free from misinformation or rumors. The results reveal that 100% of the reference links of the 100 samples are reliable, which implies high reliability of the reference links in FELM overall.

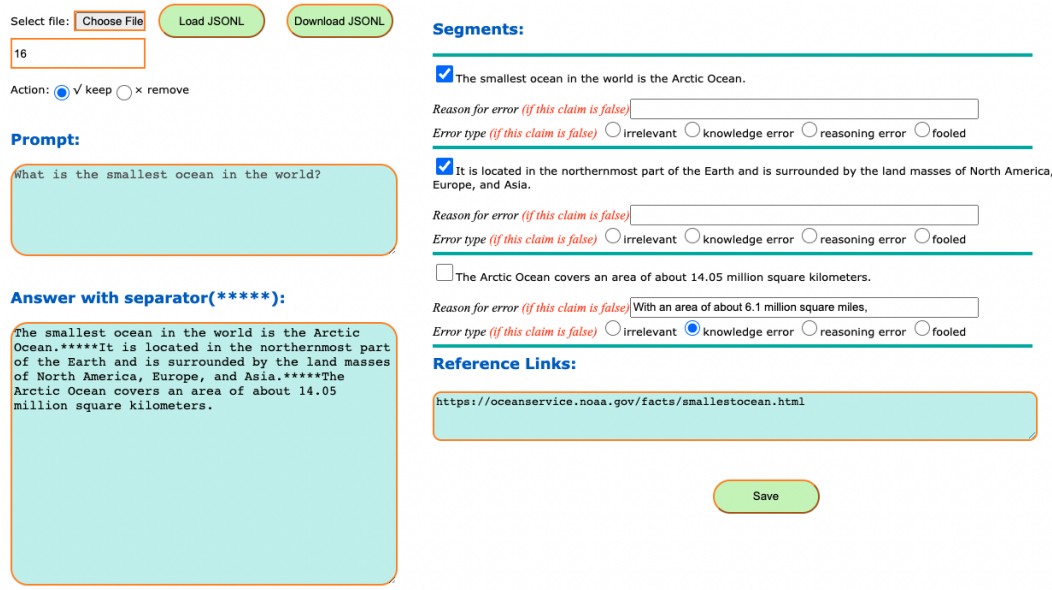

Figure 6: Annotation Page

**Safety and validity evaluation:** In order to evaluate the safety and overall quality of our GPT-generated responses, we engage the services of Amazon MTurk workers to meticulously evaluate them. Our assessment encompasses two pivotal aspects: Safety and Validity. Safety pertains to ensuring that the responses are devoid of any harassment, sexual, or violent content. On the other hand, Validity centers around confirming that the responses are complete and meaningful to the given prompt no matter whether they are correct or incorrect. We opt for a randomized selection of 100 samples, with each sample being reviewed by three distinct workers for annotation. The workers are paid 0.3 USD for each annotation. The outcomes reveal that the entirety of the 100 responses adhere to safety and validity standards.

# D  Experimental Details and Prompts

**Setup:** In our experiments, we use greedy decoding (temperature=0) to obtain the results. And we use `gpt-3.5-turbo-0301` and `gpt-4-0314` throughout the experiment. We established the maximum token limit at 1500 for claim extraction tasks and 100 for factuality detection tasks. For retrieval-augmented methods that use reference documents, we divided the retrieved documents into 512-token chunks and selected the most relevant chunk using the BM25 algorithm. In cases where there were multiple reference links, we concatenated the retrieved chunks.

**Prompts:** In the following tables, we present the prompts used in our experiments to evaluate the factuality assessment performance of ChatGPT and GPT-4 on world knowledge and writing/recommendation domain. These prompts encompass those used to extract claims from text segments which are shown at Table 5, as well as those used to evaluate the factuality of claims or segments which are shown at Table 6, 7, 8, and 9. We utilized the same extraction and factuality determination prompts for both the world knowledge and writing/recommendation domains, as the response formats are similar in these two domains. This approach allowed us to maintain consistency across both domains, which is important for reliable comparison of the performance of the two models. We use the exact wording of instructions here in our experiments.

**Cost:** We spend a total of 0.5 USD for generating responses, and 22.04 USD evaluating factuality using ChatGPT (which includes multiple iterations of attempting prompts). We spend 132.3 USD for evaluating GPT4 evaluators.

| Prompts for extracting claims of responses |
|---|

*Prompting methods for extracting claims of responses in world knowledge, writing/recommendation domains:*
I will show you a question and a list of text segments. The text segments can be concatenated to form a complete answer to the question. Your task is to extract factual claims from each text segment.

Here is one example:
Question: Tell me about the World Happiness Report.
Segments:
1. The World Happiness Report is an annual report published by the United Nations Sustainable Development Solutions Network that ranks countries by their level of happiness or subjective well-being.
2. The report aims to provide policymakers with information and analysis to help them make informed decisions about promoting happiness and well-being in their countries.

Below are your outputs:
Answer:
Segment 1:
Claim 1. The World Happiness Report is an annual report.
Claim 2. The World Happiness Report is published by the United Nations Sustainable Development Solutions Network.
Claim 3. The World Happiness Report ranks countries by their level of happiness or subjective well-being.

Segment 2:
Claim 1. The World Happiness Report aims to provide policymakers with information and analysis.
Claim 2. The World Happiness Report aims to help policymakers make informed decisions.
Claim 3. The World Happiness Report aims to help policymakers promote happiness and well-being in their countries.

Below are my inputs:

---

*Prompting methods for extracting claims of responses in science/tech domain:*
I will show you a question and a list of text segments. The text segments can be concatenated to form a complete answer to the question. Your task is to extract factual claims from each text segment.

Here is one example:
Question: What is the diffusion model in computer science?
Segments:
1. In computer science, the diffusion model is a mathematical model used to simulate the spread of information or data through a network or system.
2. It is often used to study phenomena such as the spread of viruses, the adoption of new technologies, or the dissemination of information in social networks.

Below are your outputs:
Answer:
Segment 1:
Claim 1. The diffusion model is a mathematical model.
Claim 2. The diffusion model is used in computer science.
Claim 3. The diffusion model is used to simulate the spread of information or data through a network or system.

Segment 2:
Claim 1. The diffusion model is often used to study the spread of viruses.
Claim 2. The diffusion model is often used to study the adoption of new technologies.
Claim 3. The diffusion model is often used to study the dissemination of information in social networks.

Below are my inputs:

Table 5: A one-shot prompting example to extract claims for the response segments. We use the exact wording of instructions here in our experiments.

| Vanilla prompts for factuality detection in world knowledge and writing/recommendation domains |
| --- |

*Segment-based Vanilla Prompting for world knowledge and writing/recommendation:*
I will show you a question and a list of text segments. All the segments can be concatenated to form a complete answer to the question. Your task is to assess whether each text segment contains factual errors or not.
Please generate using the following format:
Answer: List the ids of the segments with errors (separated by commas). Please only output the ids, no more details. If all the segments are correct, output "ALL_CORRECT".

Here is one example:
Question: What is the total number of nuclear power plants worldwide?
Segments:
1. there were a total of 440 operating nuclear power reactors in the world, with a total installed capacity of over 390 gigawatts (GW).
2. These reactors are located in 30 countries around the world, with the highest number of reactors in the United States, followed by France, China, Japan, and Russia.

Below are your outputs:
Answer: 1,2
It means segment 1,2 contain errors.

Below are my inputs:

*Claim-based Vanilla Prompting for world knowledge and writing/recommendation:*
I will show you a question and a list of claims. All the claims are extracted from an answer to the question. Your task is to assess whether each claim contains factual errors or not.
Please generate using the following format:
Answer: List the ids of the claims with errors (separated by commas). Please only output the ids, no more details. If all the claims are correct, output "ALL_CORRECT".

Here is one example:
Question: What is the total number of nuclear power plants worldwide?
Claims:
1. There were 440 operating nuclear power reactors in the world.
2. The total installed capacity of these reactors was over 390 gigawatts (GW).
3. The reactors are located in 30 countries around the world.
4. The highest number of reactors is in the United States.
5. France has the second-highest number of reactors.
6. China has a significant number of reactors.
7. Japan has a significant number of reactors.
8. Russia has a significant number of reactors.

Below are your outputs:
Answer: 1,2,3
It means claim 1,2,3 contain errors.

Below are my inputs:

Table 6: Evaluation prompts for one-shot vanilla methods on both segment-based and claim-based settings. We use the exact wording of instructions here in our experiments.

**Chain-of-Thought prompts for factuality detection in world knowledge and writing/recommendation domains**

*Segment-based Chain-of-Thought Prompting for world knowledge and writing/recommendation:*
I will show you a question and a list of text segments. All the segments can be concatenated to form a complete answer to the question. Your task is to assess whether each text segment contains factual errors or not.
Please generate using the following format:
Thought: Your reasoning process for the segments with errors. If all the segments are correct, output nothing.
Answer: List the ids of the segments with errors (separated by commas). Please only output the ids, no more details. If all the segments are correct, output "ALL_CORRECT".

Here is one example:
Question: What is the total number of nuclear power plants worldwide?
Segments:
1. there were a total of 440 operating nuclear power reactors in the world, with a total installed capacity of over 390 gigawatts (GW).
2. These reactors are located in 30 countries around the world, with the highest number of reactors in the United States, followed by France, China, Japan, and Russia.

Below are your outputs:
Thought: For segment 1, there are only 410 operable power reactors in the world, not 440. And the total installed capacity of these reactors was only 368.6 GW, not 390. For Segment 2, the reactors are located in 32 countries around the world, not 30.
Answer: 1,2
It means segment 1,2 contain errors.

Below are my inputs:

*Claim-based Chain-of-Thought Prompting for world knowledge and writing/recommendation :*
I will show you a question and a list of claims. All the claims are extracted from an answer to the question. Your task is to assess whether each claim contains factual errors or not.
Please generate using the following format: Thought: Your reasoning process for the claims with errors. If all the claims are correct, output nothing. Answer: List the ids of the claims with errors (separated by commas). Please only output the ids, no more details. If all the claims are correct, output "ALL_CORRECT".

Here is one example:
Question: What is the total number of nuclear power plants worldwide?
Claims:
1. There were 440 operating nuclear power reactors in the world.
2. The total installed capacity of these reactors was over 390 gigawatts (GW).
3. The reactors are located in 30 countries around the world.
4. The highest number of reactors is in the United States.
5. France has the second-highest number of reactors.
6. China has a significant number of reactors.
7. Japan has a significant number of reactors.
8. Russia has a significant number of reactors.

Below are your outputs:
Thought: For claim 1, there are only 410 operable power reactors in the world, not 440. For claim 2, The total installed capacity of these reactors was only 368.6 GW., not 390. For claim 3, the reactors are located in 32 countries around the world, not 30.
Answer: 1,2,3
It means claim 1,2 and 3 contain errors.

Below are my inputs:

Table 7: Evaluation prompts for one-shot chain-of-thought methods on both segment-based and claim-based settings. We use the exact wording of instructions here in our experiments.

| Retrieval-augmented (link) prompts for factuality detection in world knowledge and writing/recommendation domains |
| --- |

*Segment-based Retrieval Method with reference links for world knowledge and writing/recommendation:*
I will show you a question, a list of text segments, and reference links. All the segments can be concatenated to form a complete answer to the question. Your task is to assess whether each text segment contains factual errors or not with the help of the reference links.
Please generate using the following format: Answer: List the ids of the segments with errors (separated by commas). Please only output the ids, no more details. If all the segments are correct, output "ALL_CORRECT".

Here is one example:
Question: What is the total number of nuclear power plants worldwide?
Segments:
1. there were a total of 440 operating nuclear power reactors in the world, with a total installed capacity of over 390 gigawatts (GW).
2. These reactors are located in 30 countries around the world, with the highest number of reactors in the United States, followed by France, China, Japan, and Russia.
Reference Links:
https://en.wikipedia.org/wiki/Nuclear_power_by_country, https://en.wikipedia.org/wiki/List_of_commercial_nuclear_reactors

Below are your outputs:
Answer: 1,2
It means segment 1,2 contain errors.

Below are my inputs:

*Claim-based Retrieval Method with reference links for world knowledge and writing/recommendation:*
I will show you a question, a list of claims, and reference links relevant to the question and claims. All the claims are extracted from an answer to the question. Your task is to assess whether each claim contains factual errors or not with the help of the reference links.
Please generate using the following format: Answer: List the ids of the claims with errors (separated by commas). Please only output the ids, no more details. If all the claims are correct, output "ALL_CORRECT".

Here is one example: Question: What is the total number of nuclear power plants worldwide? Claims: 1. There were 440 operating nuclear power reactors in the world.
2. The total installed capacity of these reactors was over 390 gigawatts (GW).
3. The reactors are located in 30 countries around the world.
4. The highest number of reactors is in the United States.
5. France has the second-highest number of reactors.
6. China has a significant number of reactors.
7. Japan has a significant number of reactors.
8. Russia has a significant number of reactors.
Reference Links:
https://en.wikipedia.org/wiki/Nuclear_power_by_country, https://en.wikipedia.org/wiki/List_of_commercial_nuclear_reactors

Below are your outputs:
Answer: 1,2,3
It means claim 1,2 and 3 contain errors.

Below are my inputs:

Table 8: Evaluation prompts for one-shot retrieval-augmented methods with reference links on both segment-based and claim-based settings. We use the exact wording of instructions here in our experiments.

**Retrieval-augmented (doc) prompts for factuality detection in world knowledge and writing/recommendation domains**

*Segment-based Retrieval Method with reference doc for world knowledge and writing/recommendation:*
I will show you a question, a list of text segments, and a reference doc. All the segments can be concatenated to form a complete answer to the question. Your task is to assess whether each text segment contains factual errors or not with the help of the reference doc.
Please generate using the following format:
Answer: List the ids of the segments with errors (separated by commas). Please only output the ids, no more details. If all the segments are correct, output "ALL_CORRECT".

Here is one example:
Question: What is the total number of nuclear power plants worldwide?
Segments:
1. there were a total of 440 operating nuclear power reactors in the world, with a total installed capacity of over 390 gigawatts (GW).
2. These reactors are located in 30 countries around the world, with the highest number of reactors in the United States, followed by France, China, Japan, and Russia.
Reference doc:
Nuclear power plants operate in 32 countries and generate about a tenth of the world's electricity.[1] Most are in Europe, North America, East Asia and South Asia. The United States is the largest producer of nuclear power, while France has the largest share of electricity generated by nuclear power, at about 70%.[2] China has the fastest growing nuclear power programme with 16 new reactors under construction, followed by India, which has 8 under construction.[3]. As of May 2023, there are 410 operable power reactors in the world, with a combined electrical capacity of 368.6 GW.

Below are your outputs:
Answer: 1,2
It means segment 1,2 contain errors.

Below are my inputs:

*Claim-based Retrieval Method with reference doc for world knowledge and writing/recommendation:*
I will show you a question, a list of claims, and a reference doc relevant to the question and claims. All the claims are extracted from an answer to the question. Your task is to assess whether each claim contains factual errors or not with the help of the reference doc.
Please generate using the following format: Answer: List the ids of the claims with errors (separated by commas). Please only output the ids, no more details. If all the claims are correct, output "ALL_CORRECT".

Here is one example:
Question: What is the total number of nuclear power plants worldwide?
Claims:
1. There were 440 operating nuclear power reactors in the world.
2. The total installed capacity of these reactors was over 390 gigawatts (GW).
3. The reactors are located in 30 countries around the world.
4. The highest number of reactors is in the United States.
5. France has the second-highest number of reactors.
6. China has a significant number of reactors.
7. Japan has a significant number of reactors.
8. Russia has a significant number of reactors.
Reference doc:
Nuclear power plants operate in 32 countries and generate about a tenth of the world's electricity.[1] Most are in Europe, North America, East Asia and South Asia. The United States is the largest producer of nuclear power, while France has the largest share of electricity generated by nuclear power, at about 70%.[2] China has the fastest growing nuclear power programme with 16 new reactors under construction, followed by India, which has 8 under construction.[3]. As of May 2023, there are 410 operable power reactors in the world, with a combined electrical capacity of 368.6 GW.

Below are your outputs:
Answer: 1,2,3
It means claim 1,2 and 3 contain errors.

Below are my inputs:

Table 9: Evaluation prompts for one-shot retrieval-augmented methods with reference doc on both segment-based and claim-based settings. We use the exact wording of instructions here in our experiments.

# E  Additional Results

We report the balanced accuracy of all the evaluators on both the segment level and the response level under all the settings in §4 at Table 10.

| Method | | All | | WK | | Sci/Tech | | Writing/Rec | | Math | | Reasoning | |
|---|---|---|---|---|---|---|---|---|---|---|---|---|---|
| | | seg. | resp. | seg. | resp. | seg. | resp. | seg. | resp. | seg. | resp. | seg. | resp. |
| **Vicuna-33B** | | | | | | | | | | | | | |
| Segment | Vanilla | 50.6 | 53.2 | 53.4 | 57.3 | 47.7 | 47.2 | 50.9 | 57.6 | 51.9 | 55.5 | 52.4 | 54.6 |
| | Cot | 49.0 | 49.1 | 51.6 | 55.7 | 44.4 | 44.0 | 47.5 | 50.8 | 51.0 | 49.2 | 50.6 | 54.9 |
| | Link | 48.5 | 51.7 | 50.3 | 53.4 | 45.4 | 45.3 | 46.8 | 55.2 | – | – | – | – |
| | Doc | 50.6 | 52.6 | 49.8 | 49.3 | 49.6 | 47.3 | 50.6 | 58.8 | – | – | – | – |
| Claim | Vanilla | **56.5** | 52.9 | 55.3 | 50.6 | 49.8 | 50.0 | 63.4 | 51.7 | – | – | – | – |
| | Cot | 54.3 | 48.8 | 52.4 | 54.2 | 40.9 | 39.9 | 65.7 | 62.0 | – | – | – | – |
| | Link | 56.0 | 52.5 | 55.3 | 52.6 | 45.8 | 43.8 | 63.5 | 51.8 | – | – | – | – |
| | Doc | 56.1 | **53.5** | 55.1 | 50.4 | 48.3 | 51.6 | 62.9 | 54.7 | – | – | – | – |
| **ChatGPT** | | | | | | | | | | | | | |
| Segment | Vanilla | 49.7 | 49.8 | 50.0 | 49.3 | 47.8 | 48.3 | 49.3 | 47.0 | 50.2 | 50.1 | 50.5 | 51.3 |
| | Cot | 50.5 | 50.7 | 50.2 | 51.3 | 51.0 | 51.3 | 49.8 | 49.3 | 51.7 | 50.9 | 50.2 | 50.1 |
| | Link | 50.6 | 51.1 | 51.5 | 51.3 | 50.1 | 50.1 | 49.9 | 48.9 | – | – | – | – |
| | Doc | 53.1 | 54.4 | 52.9 | 53.6 | 51.0 | 55.1 | 59.5 | 48.9 | – | – | – | – |
| Claim | Vanilla | 50.7 | 49.3 | 51.0 | 49.5 | 52.6 | 52.1 | 47.4 | 38.6 | – | – | – | – |
| | Cot | 50.2 | 50.6 | 50.1 | 51.3 | 49.7 | 49.4 | 47.1 | 47.8 | – | – | – | – |
| | Link | 52.5 | 52.8 | 53.0 | 55.0 | 50.6 | 52.7 | 51.8 | 48.4 | – | – | – | – |
| | Doc | **55.9** | **55.4** | 55.2 | 52.6 | 58.0 | 59.5 | 57.5 | 58.5 | – | – | – | – |
| **GPT-4** | | | | | | | | | | | | | |
| Segment | Vanilla | 60.7 | 63.8 | 62.0 | 63.7 | 55.2 | 59.3 | 61.0 | 57.9 | 61.5 | 61.1 | 70.5 | 79.9 |
| | Cot | 63.7 | 66.8 | 66.5 | 69.6 | 57.1 | 61.9 | 57.5 | 59.0 | 61.6 | 61.9 | 77.7 | 82.7 |
| | Link | 65.1 | 66.3 | 66.6 | 67.7 | 58.1 | 61.2 | 65.4 | 64.8 | – | – | – | – |
| | Doc | **67.1** | **68.5** | 68.2 | 69.7 | 60.8 | 57.4 | 67.4 | 77.0 | – | – | – | – |
| Claim | Vanilla | 59.6 | 61.5 | 61.2 | 60.9 | 58.4 | 59.6 | 48.1 | 48.5 | – | – | – | – |
| | Cot | 58.3 | 63.8 | 67.7 | 67.0 | 55.6 | 62.7 | 42.4 | 38.2 | – | – | – | – |
| | Link | 63.3 | 64.2 | 63.5 | 66.3 | 57.9 | 56.3 | 54.1 | 56.7 | – | – | – | – |
| | Doc | 66.5 | 67.3 | 66.4 | 66.5 | 59.5 | 57.7 | 58.6 | 72.7 | – | – | – | – |

Table 10: Segment-level and Response-level balanced accuracy of factual error detectors powered by Vicuna-33B, ChatGPT and GPT-4 on FELM. We do not involve claim-based methods for math and reasoning domains cause it is often difficult to extract self-contained, atomic claims from these two domains. There is no reference for math and reasoning either. To compute the overall average for "Link" and "Doc", we account for the vanilla numbers for math and reasoning domains since these two methods degenerate to vanilla in this case. For claim-based method, we use segment-based numbers on math and reasoning domains to compute the overall average since claim-based method degenerates to segment-based in these domains. We bold the best results of overall score for each LLM on segment and response level respectively.

# F  Example for four error types

We give examples for the four error types described in our paper at Table 11.

# G  Results of Self-Consistency on Chain-of-Thought Prompting

In this section, we further run self-consistency (Wang et al., 2022a) that is commonly practiced as an effective way to improve chain-of-thought prompting. Specifically, we experiment with ChatGPT and sample 9 responses for each example, then majority voting among the 9 predictions is performed to obtain the final output. We show the results in Table 12. Self-consistency is able to significantly outperform the baseline Cot method, by 5.0 points on segment level and 11.6 points on response level respectively.

| Error Type | Prompt | Response | Comment | Reference |
|---|---|---|---|---|
| Knowledge error | What is the name of the first woman to receive a Ph.D. in computer science? | *Segment1:* The first woman to receive a Ph.D. in computer science was Dr. Barbara Liskov, who earned her degree from Stanford University in 1968. | *Comment 1:* Mary Kenneth Keller In 1965, Sister Mary Kenneth Keller PhD'65 became the nation's first woman to earn a PhD in computer science. She came close to being the first person ever, but the first man to earn the degree accepted his diploma at Washington University in Saint Louis earlier the very same day. -from Google | https://en.wikipedia.org/wiki/Mary_Kenneth_Keller |
| Fooled error | Should i be worried that new year's day 2023 falls on a friday the 13th? | *Segment 1:* There is no reason to be worried about the fact that New Year's Day 2023 falls on a Friday the 13th. ... | *Comment 1:* New Year's Day 2023 is January 1st, not falls on a Friday the 13th. | |
| Reasoning error | Which word doesn't belong with the others? Inch Kilogram Centimeter Yard | *Segment 1:* The word that doesn't belong with the others is Yard. *Segment 2:* The other options - inch, centimeter, and kilogram - are all units of measurement for length or weight/mass, whereas "Yard" is a unit of measurement for distance, which is not the same as length. | *Comment 1:* The word that doesn't belong with the others is "Kilogram" *Comment 2:* The other options - inch, centimeter, and yard - are all units of measurement for length or distance, whereas "Kilogram" is a unit of measurement for weight/mass, which is not the same as length. | |
| Irrelevant error | Recommend some online courses for me to learn Python. | *Segment 4:* 3. Python Crash Course by Eric Matthes | *Comment 4:* It's a book. | https://www.amazon.sg/Python-Crash-Course-Hands-Project-Based/dp/1593279280 |

Table 11: Examples of four error types. We show the prompts, ChatGPT's responses, annotators' comments and reference links provided by the annotators with the samples' corresponding error types here.

| Method | Level | Overall | WK | Sci/Tech | Wri/Rec | Math | Reasoning |
|---|---|---|---|---|---|---|---|
| base | seg. | 3.8/29.1/2.0 | 2.6/33.3/1.4 | 3.9/100.0/2.0 | –/0.0/0.0 | 13.8/29.0/9.0 | 1.3/25.0/0.7 |
| consist | seg. | 7.0/26.4/4.0 ↑3.2 | 6.3/41.7/3.4 ↑3.7 | 1.7/7.1/1.0 | 0.7/50.0/0.4 ↑0.7 | 16.9/26.8/12.3 ↑3.1 | 2.6/40.0/1.4 ↑1.3 |
| base | resp. | 11.0/39.1/6.4 | 8.8/66.8/4.7 | 5.1/100.0/2.6 | 7.4/28.6/4.3 | 21.7/35.7/15.6 | 3.9/25.0/2.1 |
| consist | resp. | 23.1/29.4/19.0 ↑12.1 | 24.8/50.0/16.5 ↑16.0 | 19.2/20.6/17.9 ↑14.1 | 10.0/23.1/6.4 ↑2.6 | 30.7/28.8/32.8 ↑9.0 | 18.5/33.3/12.8 ↑14.6 |

Table 12: Results for self-consistency experiment for ChatGPT-Cot evaluator. Numbers are arranged according to F1/Precision/Recall. "base" and "consist" methods indicate baseline cot method and consistency-augmented cot method respectively. "seg." and "resp." levels mean segment-level result and response-level result respectively.

