# OpenReview forum: "FELM: Benchmarking Factuality Evaluation of Large Language Models"
_NeurIPS.cc/2023/Track/Datasets_and_Benchmarks — NeurIPS 2023 Datasets and Benchmarks Poster_

### Official Review · Reviewer_Bttr · 2023-07-03
**A factuality evaluation benchmark with some question marks, but happy to discuss with the authors.**

**Rating:** 6
**Confidence:** 4
**Correctness:** The claims are mostly correct and sup…
**Clarity:** Yes, some clarification and scoping c…

**Strengths:**

+ factuality evaluation benchmark across various domains
+ solid analysis with two LLMs across various approaches

**Additional Feedback:**

N/A

**Documentation:**

The documentation is mostly adequate.

**Limitations:**

Section 5 did discuss some limitations.

**Opportunities For Improvement:**

There are a few question marks that could be addressed to strengthen this work. Some might be nice-to-haves but not strictly necessary, so feel free to argue why it is a good idea not to incorporate them.

- In line 61, the authors argued that existing benchmarks mostly follow an entailment setting. This is to imply that FELM proposes a brand new setting, but it might also be possible to consider FELM as entailment-based. In the annotation stage, expert annotators present evidence passages that supports or refutes the segment as basis for annotation. In the experiments, retrieval-augmented approaches are also entailment-based, using LLMs as NLI models to judge whether the segment is supported by the retrieved text or not. So how does FELM present a non-entailment setting? In addition, [1] is an important previous work for this paper (as referenced frequently by the authors) and they argue that "factual precision as a function of a given knowledge source" (key idea 2 in the original paper). I wonder if the authors agree with that and would non-entailment factuality evaluation ever be a valuable setting.

- While the authors adopted ChatGPT and GPT-4 throughout the experiments, I wonder if it might be possible or necessary to include non-proprietary models, such as Alpaca, Falcon, Flan-T5, among others. This might potentially make the findings more reproducible. Furthermore, which model checkpoint of ChatGPT did the authors use? gpt-3.5-turbo, or one of the two timed versions (03-15 and 06-15)?

- I wonder if the five domains proposed in Section 3.2 are clear-cut or not.

> For example, why isn't "science and techonology" considered as part of "world knowledge"? I see that the "world knowledge" comes from MMLU, which provides many STEM-related questions and information.

> In addition, "reasoning" is a broad term, encompassing mathematical [2], algorithmic [3], commonsense [4], and more forms of reasoning. What kind of reasoning is specifically branded as reasoning here? How would it overlap with the "math" domain?

> While Section 3.2 spends much space talking about the five domains, in Section 3.4 it is not clear which existing resources are leveraged according to the five different domains. While the appendix might contain relevant information, it might be helpful to at least list the existing datasets the authors incorporated for each domain.

> Finally, whether the "recommendation and writing" domain is desirable as part of factuality evaluation is subject to debate. For example, since the authors leveraged "broad and open-ended questions" sourced from Quora for this domain, the generated content might just contain opinions rather than facts. Does the FELM benchmark adequately separate opinions and facts? It would be impossible to evaluate the correctness of opinions otherwise.

- About the scale of the benchmark: only 817 samples are included in the FELM benchmark. While the annotation process is potentially labor-intensive, many of the existing resources the authors leveraged present much more examples: For example, MMLU contains more than 10k examples, how are they downsampled? Why do the authors believe 817 data instances, which is selected from diverse datasets often containing much more than that, are adequate for multi-domain factuality evaluation?

- In Section 3.5, the authors propose to "segment with ChatGPT" as part of the dataset construction process. How is that step evaluated and validated? What is the granularity of segments generated by ChatGPT compared to the first sentence-level approach?

- The four error types proposed in Section 3.6 could be better supported by existing works or linguistics theory, if any. Are there errors that do not belong to any of those categories?

- What is the compensation for the expert annotators?

- Does the "reference link" approach in Section 4.1 denote that the url is directly appended into the prompt? Why do the authors expect that the url text itself, often noisy and uninformative, might be helpful to factuality evaluation?

- In Table 4 and 5, the writing/recommendation domain for segment-based approaches always show a performance of 0 across F1, precision, and recall. Given that this is a binary classification setting (it is, right?), what does these 0s suggest? The results seem a bit fishy here.

- I suggest grouping the segment-based approaches and claim-based approaches together in Tables 4 and 5. A better visual comparison might be achieved in this way. Also, maybe include self consistency as an enhanced version of CoT to see if it helps in the context of factuality evaluation?

- Maybe include a qualitative analysis in the main paper/appendix to showcase the most typical error cases and paradigms?

- For checklist 3(b), there are some hyperparameters such as temperature that could be reported to enhance interpretability. For checklist 3(d), it would be nice to disclose the monetary cost of running all these experiments with ChatGPT and GPT-4. This would better inform readers how much they should be prepared to spend when experimenting with FELM.

[1] Min, Sewon, et al. "FActScore: Fine-grained Atomic Evaluation of Factual Precision in Long Form Text Generation." arXiv preprint arXiv:2305.14251 (2023).

[2] Shi, Freda, et al. "Large language models can be easily distracted by irrelevant context." arXiv preprint arXiv:2302.00093 (2023).

[3] Wang, Heng, et al. "Can Language Models Solve Graph Problems in Natural Language?." arXiv preprint arXiv:2305.10037 (2023).

[4] Liu, Jiacheng, et al. "Generated knowledge prompting for commonsense reasoning." arXiv preprint arXiv:2110.08387 (2021).

**Relation To Prior Work:**

It is mostly adequate, but perhaps enrich the related work section from the method side of things, especially recent LLM and retrieval-based factuality evaluation approaches?

**Summary And Contributions:**

This paper proposes FELM, a new factuality evaluation benchmark. FELM mainly tackles two issues in existing works argued by the authors: 1) domain, where the authors included five domains in the FELM benchmark; 2) granularity, where the authors mainly champion a segment-level analysis. Experiments with ChatGPT and GPT-4 demonstrate that FELM presents a challenging benchmark.

---

> ### Author Response · Authors · 2023-08-20
> **Author Response 1/3**
>
> Thanks for your time and detailed comments! Due to time limitations we could only address major points, but we’ll make sure to reflect all advice in future revisions.
>
> >  In the experiments, retrieval-augmented approaches are also entailment-based, using LLMs as NLI models to judge whether the segment is supported by the retrieved text or not. So how does FELM present a non-entailment setting?
>
> We acknowledge your point. Indeed, certain evaluation settings within FELM are based on entailment. We did not mean to suggest that FELM introduces a brand new setting. We apologize if our writing led to this miscommunication, and we have modified Section 2 a bit to mitigate such confusion. However, we think FELM is partially non-entailment because (1) math and reasoning domains in FELM are non-entailment without augmented references; and (2) our vanilla evaluation settings are non-entailment and we think LLMs themselves may be able to check factuality to some extent with their inherent knowledge stored in model parameters.
>
>
> >  [1] is an ….. and they argue that "factual precision as a function of a given knowledge source" (key idea 2 in the original paper). I wonder if the authors agree with that and would non-entailment factuality evaluation ever be a valuable setting.
>
> We only partially agree with the quoted claim in [1] because they only focused on factuality on a single domain – we do not think factuality on math and reasoning domains as in FELM quite fits in their claim. And as stated above, non-entailment factuality evaluation could be valuable in some domains or when we want to assess a model's inherent knowledge.
>
> > I wonder if it might be possible or necessary to include non-proprietary models
>
> Thanks for the advice! We have run Vicuna-33B as a representative non-proprietary model and updated the paper with Vicuna results (Table 4 and Table 10). While Vicuna is able to outperform ChatGPT in terms of F1 scores, its balanced accuracy still remains around a random level, significantly underperforming GPT-4.
>
> > Which model checkpoint of ChatGPT did the authors use?
>
> We use the gpt-3.5-turbo-0301 version. For GPT4, we use gpt-4-0314. And we have updated the paper to clarify this (Line 263-264).
>
> > I wonder if the five domains proposed in Section 3.2 are clear-cut or not.
>
> While the proposed domain definitions are a bit subjective, their specific data sources are clear-cut. For example, the world knowledge domain only uses several examples from the US Foreign Policy subject in MMLU without STEM-related questions, while the science and tech domain collects some examples from STEM subjects in MMLU. Similar topic-based separation is performed for Quora as well that is used for multiple domains. We updated the paper to clarify this (Line 158 - 167).
>
> > What kind of reasoning is specifically branded as reasoning here? How would it overlap with the "math" domain?
>
> As described between Line 164 - 167, a large portion of the reasoning data is sourced from GSM8K and the rest is mainly from online sources, where we take some reasoning examples from [2].  GSM8K represents numerical reasoning while the examples from [2] encompass diverse reasoning types including spatial reasoning, temporal reasoning, physical reasoning, and commonsense reasoning as detailed in their paper. The “math” domain predominantly comes from the MATH dataset, which focuses on more challenging mathematical problems. The distinction between "reasoning" and "math" is open to various interpretations. In our view, numerical reasoning overlaps with the “math” domain at the topic level, and “math” could be considered under a broader category of “reasoning” (as mathematical reasoning). Our choice to separate them into two domains is largely influenced by [2] (please refer to their Section 2), wherein they categorize "reasoning" and "math" as distinct categories when analyzing ChatGPT failures.
>
> > While Section 3.2 spends much space talking about the five domains, in Section 3.4 it is not clear which existing resources are leveraged according to the five different domains
>
> Thanks for the feedback! We have updated the paper to move some of the data source details from Appendix to Section 3.4 (Line 156 -167).
>
> > Whether the "recommendation and writing" domain is desirable as part of factuality evaluation is subject to debate
>
> We acknowledge the inherent subjectivity involved in domain separation, and such distinctions can indeed be debated. We include “recommendation and writing” (and other domains like “science and technology”) mainly following the separation in GPT-4 report [3] (their Figure 6), where they evaluate factuality of GPT-4 in separation of several domains that include “writing” and “recommendation”.

---

> > ### Author Response · Authors · 2023-08-20
> > **Author Response 2/3**
> >
> > > … sourced from Quora for this domain, the generated content might just contain opinions rather than facts. Does the FELM benchmark adequately separate opinions and facts?
> >
> > This is a good point. During the prompt collection and response generation phases, we filter out prompt - response pairs in which the responses solely consist of opinions.  We also filter out samples in which the responses contain unsafe opinions. And we ensured that each example contained verifiable facts and did not merely consist of opinions. The remaining opinion segments are regarded as non-factual segments. During annotation, all the non-factual segments are consistently labeled as “correct”.
> >
> > > Why do the authors believe 817 data instances, which is selected from diverse datasets often containing much more than that, are adequate for multi-domain factuality evaluation?
> >
> > We acknowledge the limitation on the scale of the dataset. This is mainly due to the costly annotation process as the reviewer mentioned. We would like to first note that while FELM only has 817 examples, it breaks down into 3948 segments, providing a substantial quantity for performing segment-level analysis. Also, while benchmarks with automatic or inherent labels like MMLU usually have a lot more examples, there are also many other commonly used benchmarks with similar numbers of examples to us, such as TruthfulQA [3] with 817 examples, AlpacaEval [4] with 805 examples, and the multi-domain MT-bench [5] with 80 examples.
> >
> > > the authors propose to "segment with ChatGPT" as part of the dataset construction process. How is that step evaluated and validated? What is the granularity of segments generated by ChatGPT compared to the first sentence-level approach?
> >
> > We aim to employ ChatGPT for segmentation at a granularity similar to the sentence-level approach – we use it as an alternative mainly because the traditional tokenizer is not robust to identify boundaries at sentence-like granularity in some domains, for example, when segmenting a structured list of recommendations or a long, single-sentence math problem solutions. For ChatGPT, specifically, the authors tried different prompts and manually examined a subset of segmentation results to ensure that the segmentation yields a similar granularity to the sentence-level approach, and in the last verification step after annotation, our super annotator also verifies the overall quality of the example that ensures segmentation is reasonable – while a rigorous comparison between the granularities of ChatGPT’s segmentation and the sentence-level method is challenging and the criteria for segmentation quality can be subjective, we deem the segmentation acceptable if it can be reasonably interpreted, given that users of FELM would follow the same segmentation scheme to report segment-based accuracy.
> >
> > > The four error types proposed in Section 3.6 could be better supported by existing works or linguistics theory, if any. Are there errors that do not belong to any of those categories?
> >
> > Thanks for the suggestion! The four error types are defined and summarized by us after we manually checked a diverse set of ChatGPT’s failure cases. We acknowledge that the definition is a bit subjective. We think that all factual errors can be classified into one of the four categories – at least during our annotation, we did not encounter any errors that fell outside these designated categories.
> >
> > > What is the compensation for the expert annotators?
> >
> > In the preliminary phase, we initially hired crowd workers from the Amazon MTurk platform to annotate randomly sampled 100 samples, where they were compensated with 0.8 USD per response annotation. However, we observed that the annotation quality did not meet our expectations. As a result, we decided to abandon crowd worker annotations and involve expert annotators who are the paper authors and our close colleagues, they perform annotations voluntarily without financial compensation.
> >
> >
> > > Does the "reference link" approach in Section 4.1 denote that the url is directly appended into the prompt? Why do the authors expect that the url text itself, often noisy and uninformative, might be helpful to factuality evaluation?
> >
> >  Yes, the url is directly appended into the prompt. The fact that this approach helps is also surprising to us. We examined this setting because we found that sometimes the url consisted of critical factual information – for instance, the link, “https://en.m.wikipedia.org/wiki/War_of_the_Worlds_(2005_film)”, offers a clue about the release date of the movie "War of the Worlds."

---

> > > ### Author Response · Authors · 2023-08-20
> > > **Author Response 3/3**
> > >
> > > >  …. show a performance of 0 across F1, precision, and recall. Given that this is a binary classification setting (it is, right?), what does these 0s suggest?
> > >
> > > We apologize for any miscommunication here! These 0s suggest that the model fails to detect any factual errors correctly, thus both the precision and recall are 0, in this case, F1 is undefined and we showed 0s in the table. We have updated Table 4 to show “--” as F1 scores in these cases.
> > >
> > > > I suggest grouping the segment-based approaches and claim-based approaches together in Tables 4 and 5
> > >
> > > Thanks for the advice! We have updated the paper to group the two tables together (plese refer to Table 4).
> > >
> > > > maybe include self consistency as an enhanced version of CoT to see if it helps in the context of factuality evaluation?
> > >
> > > Thanks for the advice! We have run self consistency experiments on ChatGPT, and show the F1/R/P results below.
> > >
> > > | All| World Knowledge | Science| Writing/rec| Math| Reasoning|
> > > |--------------------|------|------|-----------|------|------|
> > > | Cot-segment_level | 2.0/1.1/29.6| 2.6/1.4/33.3 | 3.9/2.0/100 | –/0.0/0.0| 13.8/9.0/29.0| 1.3/0.7/25.0|
> > > | Cot-consistency-segment_level| 7.0/4.0/26.4 | 6.3/3.4/41.7| 1.7/1.0/7.1 | –/0.0/0.0| 16.9/12.3/26.8 | 26.1/1.4/40 |
> > > | Cot-response_level | 11.5/6.74/37.8| 8.8/4.7/66.8 | 9.8/5.1/100 | –/0.0/0.0| 21.7/15.6/35.7| 3.9/2.1/25.0|
> > > | Cot-consistency-response_level | 23.1/19.0/29.4 | 24.8/16.5/50.0| 19.2/17.9/20.6 | –/0.0/0.0| 30.7/32.8/28.8 | 18.5/12.8/33.3 |
> > >
> > > Self-consistency improves base COT significantly, with an average of 5.0 points on segment level and 11.6 points on response level in terms of F1. We have added the self-consistency results to the paper (Section 4.2 Line 300 - 303, and Appendix G).
> > >
> > > > Maybe include a qualitative analysis in the main paper/appendix to showcase the most typical error cases and paradigms?
> > >
> > > Thanks for your advice! We have updated the paper to include typical error cases in Appendix F.
> > >
> > > > For checklist 3(b), there are some hyperparameters such as temperature that could be reported to enhance interpretability. For checklist 3(d), it would be nice to disclose the monetary cost of running all these experiments with ChatGPT and GPT-4. This would better inform readers how much they should be prepared to spend when experimenting with FELM.
> > >
> > > Thanks for your advice! We have updated the paper with more details on hyperparameters (Line 556-557) and monetary cost of running all these experiments (Line 571 - 573). In short, running all the ChatGPT evaluation in Table 4 costs around 22 USD and GPT-4 evaluation in Table 4 costs around 130 USD. This is obtained from our OpenAI invoices. It is only a rough approximation since it may involve some prompt engineering cost from our trial.
> > >
> > > [1] Min S, Krishna K, Lyu X, et al. FActScore: Fine-grained Atomic Evaluation of Factual Precision in Long Form Text Generation. arXiv. 2023.
> > > [2] Borji A. A categorical archive of chatgpt failures. arXiv.  2023.
> > > [3] Lin S, Hilton J, Evans O. Truthfulqa: Measuring how models mimic human falsehoods. arXiv. 2021.
> > > [4] Li X, Zhang T, Dubois Y, et al. Alpacaeval: An automatic evaluator of instruction-following models. 2023.
> > > [5] Zheng L, Chiang W L, Sheng Y, et al. Judging LLM-as-a-judge with MT-Bench and Chatbot Arena. arXiv. 2023.

---

> > > > ### Comment · Reviewer_Bttr · 2023-08-22
> > > >
> > > > I thank the authors for their detailed response. The proposed edits would strengthen the manuscript and I have adjusted my rating.

---

### Official Review · Reviewer_jaFr · 2023-07-19
**This article describes a benchmark test called FELM for assessing the facticity of large language models. Unlike previous factuality studies that focused primarily on world knowledge (e.g., information from Wikipedia), FELM focuses on factuality across different domains, including world knowledge, mathematics, and reasoning. FELM contributes to the development and evaluation of large language models.**

**Rating:** 8
**Confidence:** 4
**Correctness:** yes
**Clarity:** yes

**Strengths:**

1. Unlike previous factual assessments that focused primarily on world knowledge (e.g., information from Wikipedia), the FELM focuses on factualness across a variety of domains, including world knowledge, math, and reasoning.
2. The benchmark uses text passages as the basis for annotation, which can help identify specific factual errors. In addition, the benchmark provides predefined types of errors and reference links to support or refute statements.
3. The performance of several large language models on FELM is evaluated, including both regular large language models and those with added retrieval mechanisms and thought chaining processes.

**Additional Feedback:**

N/A

**Documentation:**

yes

**Ethics:**

It's fine.

**Limitations:**

above

**Opportunities For Improvement:**

It would be valuable if the author could provide a multi-language version.

**Relation To Prior Work:**

yes

**Summary And Contributions:**

This article describes a benchmark test called FELM for assessing the facticity of large language models. Unlike previous factuality studies that focused primarily on world knowledge (e.g., information from Wikipedia), FELM focuses on factuality across different domains, including world knowledge, mathematics, and reasoning. Unlike previous factual studies that have focused primarily on world knowledge (e.g., information from Wikipedia), FELM focuses on factuality across different domains, including world knowledge, mathematics, and reasoning. The authors also conducted experiments to investigate the performance of several factuality evaluators based on LLMs on FELM, including regular LLMs as well as those with added retrieval mechanisms and chains of thought.The results show that, while retrieval facilitates factuality evaluation, current LLMs are far from being able to accurately detect factual errors.

---

> ### Author Response · Authors · 2023-08-20
>
> Thank you for your encouraging comments and the advice on multilingual extension! We will consider creating a multilingual version of FELM in the future.

---

### Official Review · Reviewer_jaKw · 2023-07-21
**Review for FELM**

**Rating:** 7
**Confidence:** 4
**Correctness:** The dataset is constructed in a sound…
**Clarity:** The paper is well-written.

**Strengths:**

1. The proposed dataset is valuable for evaluating the ability of LLMs and promoting further research. FELM covers various domains and involves human annotators to provide fine-grained labels and references.
2. The authors provide the details of the data generation process, from prompts collection, response generation, and segmentation, to annotation and quality checking. Besides, the authors also provide the design principle and intuition of FELM, which might be useful for researchers to follow their study.
3. Based on the dataset, the authors conduct some experiments with ChatGPT and GPT-4, providing discussions on how to further improved the ability of LLMs as factuality evaluators.
4. The paper is well-written and easy to follow.


**Additional Feedback:**

Please refer to the above comments

**Documentation:**

The authors public the dataset and provide enough details.

**Limitations:**

The authors provide limitations in this submission.

**Opportunities For Improvement:**

1. The empirical comparison between FELM and related works is not provided in this submission. It would be interesting to see how ChatGPT/GPT-4 performs on other benchmark datasets, such as the one provided in *On Faithfulness and Factuality in Abstractive Summarization*. Such a comparison can be helpful to better understand the difficulty and meaningfulness of the proposed FELM compared to the existing works, and show the generalization of proposed ideas of prompt engineering.
2. The experiments show that CoT prompting method promotes GPT-4 but fails to help ChatGPT for all the settings. One suggestion is to change the CoT prompts for further confirming such a conclusion, such as splitting it into subtasks.


**Relation To Prior Work:**

The authors provide discussions on the comparison between the proposed dataset and previous works.

**Summary And Contributions:**

In this submission, the authors propose a benchmark dataset, called FELM, to evaluate the ability of LLMs as factuality evaluators, consisting of world knowledge, science and technology, math, writing and recommendation, and reasoning. The proposed FELM contains 817 samples and 3948 segments, and each segment is annotated factuality labels, error reasons, error types, and references by human annotators. Experiments conducted on FELM show the proposed dataset is a challenging benchmark for both ChatGPT and GPT-4, and some techniques such as CoT prompts and external links/docs might help LLMs achieve better performance.

---

> ### Author Response · Authors · 2023-08-20
>
> Thanks for your encouraging comments! We address your comments below.
>
>
> > The empirical comparison between FELM and related works is not provided in this submission. It would be interesting to see how ChatGPT/GPT-4 performs on other benchmark datasets, such as the one provided in On Faithfulness and Factuality in Abstractive Summarization.
>
> Thanks for the advice! The dataset mentioned by the reviewer is a summarization factualness dataset and often referred to as XSumFaith. We note that there are already some works reporting ChatGPT/GPT-4 performance on detecting factual errors in abstractive summarization including XSumFaith, here we directly cite their numbers for reference – for example, ChatGPT (zero-shot) showed around 60%-70% balanced accuracy in diverse summarization factualness datasets [1], their Table 8 showed the specific numbers on XSumFaith. [2] reported 67% - 70% balanced accuracy on XSumFaith for both ChatGPT and GPT-4 (their Table 2). On simpler datasets like SummEval [3], ChatGPT and GPT-4 are able to make an over 80% balanced accuracy as demonstrated in [2]. We observe that these numbers are generally higher than the ones on FELM, implying that open-ended factual error detection is harder than detecting factual errors in summaries given the corresponding documents. We have updated the paper to cite these numbers and discussed the difficulty of FELM compared to previous summarization datasets (Line 282-289).
>
>
> > The experiments show that CoT prompting method promotes GPT-4 but fails to help ChatGPT for all the settings. One suggestion is to change the CoT prompts for further confirming such a conclusion, such as splitting it into subtasks.
>
> Thanks for the suggestion! In the next revision, we will try more advanced COT prompts such as task decomposition as suggested to further confirm the conclusion on COT prompting. In the meanwhile, we would like to note that we run self-consistency on top of COT prompting and add the results to Appendix G – self-consistency is able to boost ChatGPT’s COT performance greatly and the resulting numbers surpass the vanilla prompting ones of ChatGPT.
>
> [1] Tang L, Goyal T, Fabbri A R, et al. Understanding factual errors in summarization: Errors, summarizers, datasets, error detectors. ACL. 2023.
>
> [2] Chen S, Gao S, He J. Evaluating Factual Consistency of Summaries with Large Language Models. arXiv. 2023.
>
> [3] Fabbri A R, Kryściński W, McCann B, et al. Summeval: Re-evaluating summarization evaluation. ACL. 2021.
>
> [4] Wang X, Wei J, Schuurmans D, et al. Self-consistency improves chain of thought reasoning in language models. arXiv. 2022.

---

> > ### Comment · Reviewer_jaKw · 2023-08-29
> > **Reply**
> >
> > Thanks to the authors for a detailed response that addressed my questions.

---

### Official Review · Reviewer_x7TP · 2023-07-23
**An important advancement in evaluating LLMs factuality; however, the evaluation study lacks rigor, and the benchmark lacks ethical and safety considerations, besides having a small size.**

**Rating:** 6
**Confidence:** 3
**Clarity:** The paper is well-written and easy to…

**Strengths:**

- The study focuses on a crucial aspect needed by the NLP community, evaluating factuality in LLMs, which is essential given the challenges that hallucination can pose.
- The collected benchmark represents a significant step forward in providing improved evaluation metrics for assessing the factuality of LLMs.

**Additional Feedback:**

No

**Correctness:**

The collection process appears sensible, but the study lacks a thorough examination of potential safety concerns that could arise from these LLMs. It would have been beneficial if the authors had discussed the associated risks and explored the possibility of implementing filtering mechanisms to mitigate potential issues.

**Documentation:**

There is no documentation about ethical and responsible use.

**Ethics:**

My concerns about ethics are only minor; however, I suggest that the authors add a section to discuss this matter.

**Limitations:**

The authors have overlooked addressing the limitations and potential negative societal impact of the work. The absence of safety measures or investigation of the collected benchmark raises concerns that have not been adequately addressed. Read above for the rest of the limitations.

**Opportunities For Improvement:**

- One of the main contributions of the paper is the collection of fine-grained factuality errors. However, the study missed an opportunity to leverage this collected data effectively. It could be highly beneficial for the study to showcase how fine-grained annotation can improve model performance through training a reward model.
- The related work section lacks important references to evaluation benchmarks for LLMs, specifically in the Dialogue domain, such as BEGIN (Dziri et al., 2022) [1], which evaluates factuality based on knowledge snippets from Wikipedia.
- The reliance on annotators to find external evidence raises valid concerns about the trustworthiness of knowledge sources and the possibility of misinformation or factual inaccuracies. Rigorous evaluation of annotator performance is necessary.
- While the paper uses F1 and balanced classification accuracy as evaluation metrics, it overlooks the exploration of other automatic metrics, like hallucination critics, that could enhance the assessment of factuality, as demonstrated by Dziri et al. [2] and FeQA by Durmus et al [3].
- Evaluating ChatGPT's own generation might produce misleading results, considering the model's propensity for hallucinating information.
- Human validation to assess the factuality correctness of generated LLM responses is missing, which could provide valuable insights into model performance.
- The reported average increase in F1 through ChatGPT's retrieval-augmented reference document method is expected, as providing evidence documents to the model boosts its performance in detecting factual errors.
- No safety measures or additional human annotation to validate the quality of the collected benchmark have been conducted, which leaves potential concerns unaddressed.
- The benchmark's size is relatively small, and a larger dataset might strengthen the study's findings and generalizability.


References:

[1] Dziri N, Rashkin H, Linzen T, Reitter D. Evaluating attribution in dialogue systems: The BEGIN benchmark. Transactions of the Association for Computational Linguistics. 2022 Sep 19;10:1066-83.
[2] Dziri N, Kamalloo E, Milton S, Zaiane O, Yu M, Ponti EM, Reddy S. Faithdial: A faithful benchmark for information-seeking dialogue. Transactions of the Association for Computational Linguistics. 2022 Dec 23;10:1473-90.
[3] Durmus E, He H, Diab M. FEQA: A question answering evaluation framework for faithfulness assessment in abstractive summarization. arXiv preprint arXiv:2005.03754. 2020 May 7.

**Relation To Prior Work:**

Some important related works are missing as explained in the limitations.

**Summary And Contributions:**

The paper presents FELM, a benchmark specifically designed for evaluating the factuality of LLMs consisting of a total of 817 samples. FELM aims to assess the accuracy of LLMs in determining factuality across diverse domains, ranging from world knowledge to mathematical reasoning.

The construction of FELM involves a four-step process. Firstly, prompts are gathered from various sources such as online platforms like Quora, standard benchmarks such as MMLU  to provide a diverse set of contexts for evaluation. Next, responses generated by LLMs, specifically ChatGPT, are collected and segmented into fine-grained text spans. Each segment is then annotated for factuality by human annotators, who also provide additional information on error type, error reason, and reference links used to make the judgment.

The findings of the evaluation highlight a challenge in factual error detection for LLMs. Despite being powerful models, both ChatGPT and GPT-4 demonstrate limitations in accurately determining factuality. The study underscores the need for external tools and techniques to improve the performance of LLMs in factuality evaluation.

---

> ### Author Response · Authors · 2023-08-20
> **Author Response 1/2**
>
> Thanks for your time and comments! Due to time limitations we could only address major points, but we’ll make sure to reflect all advice in future revisions.
>
>
> > It could be highly beneficial for the study to showcase how fine-grained annotation can improve model performance through training a reward model.
>
> This is a great point! We agree that the fine-grained annotation could be leveraged in many ways in addition to evaluation, such as training a reward model to improve model performance as the reviewer mentioned. However, the purpose of this paper is to establish a benchmark to evaluate factuality evaluators, which we believe is an appropriate contribution fit into the NeurIPS datasets and benchmarks track. Other usage of the data besides evaluation is beyond the scope of this paper and we leave it as future work.
>
> > The related work section lacks important references to evaluation benchmarks for LLMs, specifically in the Dialogue domain.
>
> Thanks for pointing out related work! We have updated the related work section (Line 68-75) to include more factuality benchmarks in dialogue domains as suggested.
> > The reliance on annotators to find external evidence raises valid concerns about the trustworthiness of knowledge sources.
>
> Thanks for the advice! We would like to first note that we have conducted and scrutinized the annotation process carefully as described in Section 3.6 – the annotations are from 6 expert annotators who are NLP researchers, each of the sample is annotated by two annotators and a super annotator verifies all the annotations in the last step. However, we understand the reviewer’ concerns and we do share the point that the knowledge sources themselves may be unreliable. To address this potential issue, we perform another human evaluation of the reference quality. Specifically, we randomly select 100 samples from FELM, each accompanied by reference links, then we ask the paper authors to assess the Reliability of the reference, which measures whether the linked content is free from misinformation or rumors. The results reveal that **100%** of the reference links of the 100 samples are reliable, which implies high reliability of the reference links in FELM overall. We have updated the paper and added this verification experiment to Section 3.6 (Line 231-237) and Appendix C.
>
> > While the paper uses F1 and balanced classification accuracy as evaluation metrics, it overlooks the exploration of other automatic metrics, like hallucination critics, that could enhance the assessment of factuality, as demonstrated by Dziri et al. and FeQA by Durmus et al.
>
> There may be a misunderstanding on the metrics, and we apologize if our paper did not provide a clear explanation of this. We would like to clarify that the hallucination critics as in Dziri et al. and FeQA serve as metrics for evaluating factualness of generated text, while FELM aims to evaluate **factuality detectors rather than text generation**  – The factuality detectors act as classification systems and are essentially predicting binary labels for the segments and we check whether they match the binary gold labels – we believe that accuracy and the F1 score of identifying factual errors are very standard and well-defined metrics without obvious concerns in such a typical classification scenario, and the mentioned hallucination critics metrics are for different purposes and cannot be applied here.
>
> > Evaluating ChatGPT's own generation might produce misleading results
>
> This is a good point. In the original submission, we evaluated both ChatGPT and GPT-4, in the rebuttal we have updated the paper to add results from another strong open-source model, Vicuna-33B (Please refer to Table 4 and Table 10). While Vicuna is able to outperform ChatGPT in terms of F1 scores, its balanced accuracy still remains around a random level, significantly underperforming GPT-4.

---

> > ### Author Response · Authors · 2023-08-20
> > **Author Response 2/2**
> >
> > > No safety measures or additional human annotation to validate the quality of the collected benchmark have been conducted, which leaves potential concerns unaddressed.
> >
> > Thank you for your suggestion! We conduct an additional evaluation to assess the safety and validity of FELM. Safety measures whether the examples contain toxic content, while validity examines whether the responses are complete and meaningful to the query. Specifically, we randomly select 100 samples of FELM, and hire annotators from Amazon MTurk, with 3 annotators assigned to each response. There are a total of 270 participants. We take the majority vote among 3 annotators for each sample. We found that **ALL** the 100 samples are assessed as valid and safe, without any toxic content. We have added this human verification results to section 3.6 (Line 231-237) and Appendix C.
> >
> > Regarding the quality overall, we would like to highlight that the authors already performed a rigorous verification and modification process when we created FELM as described in Line 228-231, and any low-quality or toxic samples were removed in the verification step. Therefore, we consider the resulted FELM dataset high-quality and safe overall as demonstrated in the added human verification.
> >
> > > The benchmark's size is relatively small, and a larger dataset might strengthen the study's findings and generalizability.
> >
> > We acknowledge this limitation, which largely stems from our costly annotation process. It is worth highlighting that, although FELM comprises 817 samples, it breaks down into 3948 segments, providing a substantial quantity for evaluating segment-level accuracy. Additionally, we would like to note that many other commonly used LLM benchmarks share similar limitations due to cost considerations, for example, the standard LLM alignment benchmarks AlpacaEval [1] and MT-Bench [2] contain 805 and 80 samples respectively. In the future, we will consider adding more samples to FELM if the resource is allowed.
> >
> > [1] Li X, Zhang T, Dubois Y, et al. Alpacaeval: An automatic evaluator of instruction-following models. 2023.
> >
> > [2] Zheng L, Chiang W L, Sheng Y, et al. Judging LLM-as-a-judge with MT-Bench and Chatbot Arena. arXiv. 2023.

---

> > > ### Comment · Reviewer_x7TP · 2023-08-27
> > >
> > > I thank the authors for the explanation. They have addressed some of my concerns. As such, I have raised my score to 6.

---

> > > > ### Author Response · Authors · 2023-08-27
> > > >
> > > > We are glad to hear that our response addressed some of your concerns, and thank you for considering raising the review score! However, the original review rating seems unchanged as we leave this comment, have you updated the review already?

---

### Official Review · Reviewer_hoDi · 2023-07-27
**A benchmark for evaluating factualness evaluators**

**Rating:** 6
**Confidence:** 3

**Strengths:**

-	An interesting and important research problem, i.e., evaluating the factualness of LLMs, is studied.
-	Carefully designed benchmarks covering diverse error types and graded labels.
-	Results show that the ChatGPT and GPT-4 based factualness evaluation is far from optimal and factual error detection is very challenging.


**Additional Feedback:**

None

**Clarity:**

- Yes, but it would be great if the experimental design can be well discussed and introduced.


**Correctness:**

- Yes, but it would be great if more evaluation scenarios can be discussed.


**Documentation:**

Yes

**Limitations:**

Yes

**Opportunities For Improvement:**

-	The benchmark is supposed to evaluate “factual error detectors”, however, evaluating the factualness of LLM would be more important and interesting.
-	As the benchmark is created based on ChatGPT, it is unknown whether the benchmark is available for evaluating other LLMs. As the data annotation is costly, it would be great if the evaluation can be applied to other LLMs, otherwise, the impact and technical contribution of this benchmark is quite limited. The authors should at least discuss how can this dataset be used for this kind of evaluation if it is applicable. Section 4.1 is a little bit hard to read. A clear flowchart is expected.
-	The number of samples is relatively small.



**Relation To Prior Work:**

Yes.

**Summary And Contributions:**

-	A benchmark named FELM for evaluating factualness evaluators.
-	Factuality evaluation over diverse domains, from world knowledge to math and reasoning
-	Interesting conclusions about the factuality of LLMs are made

---

> ### Author Response · Authors · 2023-08-20
>
> Thanks for your time and comments! We address your comments below.
>
>
> > The benchmark is supposed to evaluate “factual error detectors”, however, evaluating the factualness of LLM would be more important and interesting.
>
> Thanks for the comment! We agree that assessing the factualness of LLMs is very important. However, as factual error detection gains momentum as a crucial research direction for LLM applicability, we think that FELM stands as a timely benchmark in this direction to fill the gap of evaluation. While it operates as a benchmark for factuality detectors, its role is complementary to the direct assessment of LLM factualness. Both aspects, in orthogonal, play a vital role in ensuring the reliability and trustworthiness of LLMs.
>
> > As the benchmark is created based on ChatGPT, it is unknown whether the benchmark is available for evaluating other LLMs.
>
> This is a good point. We collected ChatGPT responses for the dataset because ChatGPT is the most commonly used LLM nowadays. Regarding the generality of FELM, on the one hand, we first show that our benchmark can be used to evaluate factuality detectors that are not based on GPTs – we have updated the paper to add Vicuna-33B results in Table 4. On the other hand, there may exist a potential performance gap when using factuality detectors tested on FELM to detect factual errors of generation from other LLMs. We have added discussion on this to the end of the paper (Line 324-329), we copy the added discussion here:
>
> “The responses in FELM are collected solely from ChatGPT, thus there may exist a potential performance gap when using factuality detectors tested on FELM to detect factual errors of generation from other LLMs. Such a performance gap is not trivial to study without factual annotations of responses from other LLMs. One possible remedy to mitigate this issue is to annotate and add more examples to FELM generated from a diverse range of LLMs in addition to ChatGPT, we leave it as a potential future plan to improve FELM.”
>
>
> > Section 4.1 is a little bit hard to read. A clear flowchart is expected.
>
> Thanks for the advice! We have added a flowchart figure (Figure 5) to the revised submission, hopefully it will help clarify the experiment settings in Section 4.1.
>
> > Number of samples is small.
>
> We acknowledge this limitation, which largely stems from our costly annotation process. It is worth highlighting that, although FELM comprises 817 samples, it breaks down into 3948 segments, providing a substantial quantity for evaluating segment-level accuracy. Additionally, we would like to note that many other commonly used LLM benchmarks share similar limitations due to cost considerations, for example, the standard LLM alignment benchmarks AlpacaEval [1] and MT-Bench [2] contain 805 and 80 samples respectively. In the future, we will consider adding more samples to FELM if the resource is allowed.
>
> [1] Li X, Zhang T, Dubois Y, et al. Alpacaeval: An automatic evaluator of instruction-following models. 2023.
>
> [2] Zheng L, Chiang W L, Sheng Y, et al. “Judging LLM-as-a-judge with MT-Bench and Chatbot Arena”. arXiv. 2023.

---

### Author Response · Authors · 2023-08-20
**We have updated the paper according to reviewers' comments, we thank all the reviewers for their time**

We thank the reviewers for the time and comments, and we have replied to the comments of each reviewer separately in the respective thread. Also, we have updated the paper to reflect reviewers’ comments, all the updated text is in blue to differentiate from the original submission, and the main updates are:

* Additional results on Vicuna-33B in Table 4 (Reviewer hoDi, x7TP, Bttr)
* Additional human verification experiments on reference reliability and response safety in Appendix C (Reviewer x7TP)
* Additional self-consistency experiments in Appendix G (Reviewer Bttr)
* Some clarifications and details are added across the paper to reflect the reviewers’ comments

---

### Decision · Program_Chairs · 2023-09-22

**Decision:**

Accept (Poster)

**Comment:**

This paper introduces the FELM dataset to evaluate factuality of LLMs. The dataset consists of diverse domains, fine-grained labels,
Since the hallucination of LLMs became a crucial and timely problem as the increasing prevalence of recent LLMs, I believe this benchmark, along with the reported evaluation results, can contribute the improvement of language models. Majority of the reviewers' concerns have been addressed through the authors' comprehensive responses.